# Interplay between copper redox and transfer and support acidity and topology in low temperature NH$_3$-SCR

Yiqing Wu[1,7], Wenru Zhao[2,7], Sang Hyun Ahn[3], Yilin Wang[1], Eric D. Walter[1], Ying Chen[1], Miroslaw A. Derewinski[1,4], Nancy M. Washton[1], Kenneth G. Rappé[1], Yong Wang[1,5], Donghai Mei[2,6]✉, Suk Bong Hong[3]✉ & Feng Gao[1]✉

Low-temperature standard NH$_3$-SCR over copper-exchanged zeolite catalysts occurs on NH$_3$-solvated Cu-ion active sites in a quasi-homogeneous manner. As key kinetically relevant reaction steps, the reaction intermediate Cu$^{II}$(NH$_3$)$_4$ ion hydrolyzes to Cu$^{II}$(OH)(NH$_3$)$_3$ ion to gain redox activity. The Cu$^{II}$(OH)(NH$_3$)$_3$ ion also transfers between neighboring zeolite cages to form highly reactive reaction intermediates. Via *operando* electron paramagnetic resonance spectroscopy and SCR kinetic measurements and density functional theory calculations, we demonstrate here that such kinetically relevant steps become energetically more difficult with lower support Brønsted acid strength and density. Consequently, Cu/LTA displays lower Cu atomic efficiency than Cu/CHA and Cu/AEI, which can also be rationalized by considering differences in their support topology. By carrying out hydrothermal aging to eliminate support Brønsted acid sites, both Cu$^{II}$(NH$_3$)$_4$ ion hydrolysis and Cu$^{II}$(OH)(NH$_3$)$_3$ ion migration are hindered, leading to a marked decrease in Cu atomic efficiency for all catalysts.

Owing to its remarkable activity and hydrothermal stability, copper-exchanged small-pore zeolite SSZ-13 (Cu/CHA) was commercialized in 2010 as an ammonia selective catalytic reduction (NH$_3$-SCR) catalyst to eliminate NOx in diesel engine exhausts[1–3]. In recent years, other small-pore Cu-zeolites, i.e., Cu/SSZ-39 (Cu/AEI)[4–6] and high-silica Cu/LTA[7–9], have also been found to possess great SCR activity/selectivity, and even better hydrothermal stability than Cu/CHA, rendering them attractive as substitutes for Cu/CHA. Undoubtedly, because of differences in their cage architecture, understanding how the support properties influence reactivity of SCR active Cu ions is of fundamental importance. Both CHA and AEI structures are constructed by stacking double 6-membered ring units. Such units are connected via different orientations in these two materials, generating 20-hedral ([4$^{12}$6$^2$8$^6$]) *cha* and *aei* cages with the same 8-membered ring (8MR) opening (3.8 × 3.8 Å) but different cage shapes and three-dimension (3D) connectivities, resepctively[10]. On the other hand, the LTA structure is built by linking small 14-hedral ([4$^6$6$^8$8$^6$]) *sod* cages to form significantly larger 26-hedral ([4$^{12}$6$^8$8$^6$]) *lta* cages that are also connected in 3D by 8MR windows of 4.1 × 4.1 Å.

Fundamental SCR mechanistic studies demonstrate that while the isolated Cu ions are the active sites in these catalysts, low-temperature (<250 °C) standard NH$_3$-SCR (4NO + 4NH$_3$ + O$_2$ = 4N$_2$ + 6H$_2$O) occurs

[1]Institute for Integrated Catalysis, Pacific Northwest National Laboratory, Richland, WA 99354, US. [2]School of Materials Science and Engineering, Tiangong University, Tianjin 300387, China. [3]Center for Ordered Nanoporous Materials Synthesis, Division of Environmental Science and Engineering, POSTECH, Pohang 37673, Republic of Korea. [4]J. Haber Institute of Catalysis and Surface Chemistry, Polish Academy of Sciences, 30-239 Krakow, Poland. [5]Voiland School of Chemical Engineering and Bioengineering, Washington State University, Pullman, WA 99163, US. [6]School of Environmental Science and Engineering, Tiangong University, Tianjin 300387, China. [7]These authors contributed equally: Yiqing Wu, Wenru Zhao. ✉e-mail: dhmei@tiangong.edu.cn; sbhong@postech.ac.kr; feng.gao@pnnl.gov

on NH$_3$-solvated Cu ions within zeolite cages via the redox mechanism in a quasi-homogeneous manner[11–14]. Intercage transfer of such Cu ions is indispensable for catalytic efficiency since the formation of certain key reaction intermediates requires the participation of a pair of isolated Cu ions within the same cage[12,14–17]. As such, Cu ion mobility, in particular, the intercage transfer capacity that typically increases with increasing Cu loading, has been demonstrated to play important kinetic roles[18]. However, the Cu atomic efficiency in low-temperature SCR is further complicated by the composition (Si/Al ratio) and framework Al distribution of zeolite supports because of at least two reasons: (i) two types of SCR active isolated Cu$^{II}$ ions coexist in these catalysts: the redox-resistant Z$_2$Cu$^{II}$ and the redox-active ZCu$^{II}$OH species[3], where Z is the AlO$_4^-$ unit in zeolite supports; and (ii) residual Si-OH-Al Brønsted acid sites (BASs) always exist in Cu-zeolite catalysts and their density and distribution are influenced by the Cu exchange level and zeolite framework Al content/distribution. Zeolite BASs have been proposed to directly contribute to SCR by decomposing intermediates generated on Cu sites and then spilling over to them[15,19], or indirectly by adsorbing NH$_3$ as the so-called "NH$_3$ reservoir"[20].

Here, we elucidate the rate-controlling steps of low-temperature standard SCR over four small-pore Cu-zeolites with different zeolite supports and/or Al contents, denoted Cu/LTA, Cu/AEI, Cu/CHA-a and Cu/CHA-b (Supplementary Tables 1 and 2 and Supplementary Fig. 1). Under the current state-of-the-art for small pore zeolite synthesis, CHA can be synthesized with a wide range of Si/Al ratios, however, it is not yet the case for LTA and AEI. Therefore, we prepared CHA-a and CHA-b supports to match Si/Al ratios of our LTA and AEI supports, respectively. Regarding Cu loading, we used aqueous solution ion exchange to obtain catalysts with intermediate Cu/Al ratios to avoid the presence of multinuclear Cu moieties to ease our kinetics and EPR studies. We believe that such a catalyst preparation approach enables fair comparisons among the 4 catalysts, allowing us to understand the relationship between the Cu redox and transfer and the zeolite BAS strength and topology. We use SCR kinetics to derive Cu atomic efficiency, operando electron paragenetic resonance (EPR) spectroscopy to monitor the mobility of Cu ions, and density functional theory (DFT) calculations to address their intercage transfer and possible roles of BAS strength. The results presented in this study will provide the basis for the rational design of more efficient and robust Cu-zeolite catalysts for NH$_3$-SCR.

## Results and discussion
### Low-temperature standard NH$_3$-SCR kinetics
The standard SCR reaction results, including NOx/NH$_3$ light-off plots, the formation of side products NO$_2$ and N$_2$O, and SCR selectivity vs. temperature plots, can be found in Supplementary Fig. 2. Based on NOx conversion data collected under kinetic control and further corrected using the first-order rate expression, Arrhenius analysis (detailed in Methods) was carried out to obtain key kinetic parameters for comparison purposes. Figure 1a–c compares the pre-exponential factors (A, s$^{-1}$), apparent activation energies (Ea, kJ mol$^{-1}$) and turnover rates (TOR, s$^{-1}$) of the fresh form of the four Cu-zeolites studied here. Both A and Ea values follow the order Cu/LTA < Cu/CHA-b < Cu/CHA-a ≈ Cu/AEI. Since low-temperature SCR occurs on mobile NH$_3$-solvated Cu ions, reaction rate is governed by intrinsic kinetics, intercage transfer of Cu ions, or their combinations[12,14,21]. Previous studies demonstrated that improving Cu ion intercage transfer enhances Cu atomic efficiency[18], which is readily reflected by the increase of A and Ea values. This can be corroborated by the TOR vs. 1/T Arrhenius plots in Fig. 1c, showing that Cu atomic efficiency follows the same order as that described above. On the other hand, while Cu/CHA-a and Cu/AEI contain similar quantities of Cu ions, the two supports have different topologies and Si/Al ratios (Supplementary Table 1). The AEI support has a lower Si/Al ratio than CHA-a (ca. 10 vs. 17), rendering it capable of accommodating a higher amount of Z$_2$Cu$^{II}$ as clearly evidenced by H$_2$-

TPR results in Supplementary Fig. 3, and quantitative Cu speciation data measured with EPR (will be discussed below).

The nearly identical Cu atomic efficiency of Cu/CHA-a and Cu/AEI, therefore, indicates that support topology and Z$_2$Cu$^{II}$ vs. ZCu$^{II}$OH speciation play relatively insignificant roles in rate-controlling as opposed to Cu loading. Such a situation is readily understood: (i) CHA and AEI topologies are highly similar in terms of cage and pore opening sizes[4]; and (ii) a recent study by Hu et al.[22] demonstrated that over Cu/CHA, Z$_2$Cu$^{II}$ hydrolysis to ZCu$^{II}$OH under low-temperature SCR conditions is both thermodynamically and kinetically favorable, leading to their indistinguishable atomic efficiency. Following this rationale, the lower atomic efficiency for Cu/CHA-b as compared to Cu/CHA-a and Cu/AEI is readily attributed to its lower overall Cu ion content (rather than its Z$_2$Cu$^{II}$ vs. ZCu$^{II}$OH speciation). With such an active site concentration decrease, the degree of rate control from Cu ion transfer increases, but that from the intrinsic redox chemistry decreases[18]. The catalytic behavior for Cu/LTA, however, is more difficult to interpret because it possesses the highest Cu ion content (Supplementary Table 1) but displays the lowest SCR rate. A few possibilities can be considered to explain the low Cu atomic efficiency of Cu/LTA: (i) its support interacts strongly with Cu ions and restricts their mobility, i.e., they do not transfer as efficiently as their counterparts in CHA or AEI during low-temperature SCR; (ii) Z$_2$Cu$^{II}$ hydrolysis to ZCu$^{II}$OH during low-temperature SCR over Cu/LTA is not thermodynamically or kinetically favorable; note that only ZCu$^{II}$OH was suggested to be redox active[3,22]; and (iii) in addition to Cu redox and transfer, other factors, e.g., support BAS strength and topology, also play rate-controlling roles in Cu/LTA. Using CHA-a and LTA as supports, we also synthesized two series of catalysts with identical Cu loadings via solid-state ion exchange, and then compared their SCR performance. Their Cu atomic efficiency trend fully reproduced the results shown here. The results will be published elsewhere.

### Ex situ characterization
Unlike the exchanged Cu$^{II}$ ions that adopt Z$_2$Cu$^{II}$ and ZCu$^{II}$OH configurations in zeolites, Co$^{II}$ ions only stay as Z$_2$Co$^{II}$. Therefore, Co$^{II}$ ion titration is used here to probe zeolite framework Al pairs (Al-Si-Al or Al-Si-Si-Al linkages; lattice O atoms omitted for simplicity)[13,23]. From the results in Supplementary Table 3, our CHA-a/b and AEI supports display similar Co/Al ratios of ~0.25, which suggests random framework Al distribution in these three supports[13]. In contrast, the LTA support shows a much higher Co/Al ratio of 0.39, i.e., markedly enriched framework Al pairing, expecting Cu/LTA to contain more Z$_2$Cu$^{II}$ than the other three catalysts. Another key difference between the LTA support and the other three is BAS strength. The NH$_3$-TPD curves of their proton form in Supplementary Fig. 4a show that while the high-temperature peak from H/CHA-a/b and H/AEI, assignable to NH$_3$ desorption from BASs[24–26], is centered around 460 °C, the same peak from H/LTA appears around 390 °C. We also note that the NH$_3$-TPD curves of each of the Cu-zeolites display an additional peak around 300 °C (Supplementary Fig. 4b) due to NH$_3$ desorption from Cu ions[24–27]. They also show the residual BASs with strengths following the same order as the respective supports.

We suggested above that one possible explanation to the lower SCR rate of Cu/LTA is strong Cu-support interaction that restricts Cu mobility. Therefore, we applied EPR spectroscopy to probe the interactions between Cu ions and zeolite supports in hydrated and dehydrated states. Figure 2a presents the EPR spectra for fully hydrated catalysts (i.e., Cu$^{II}$ ions stay as [Cu$^{II}$(H$_2$O)$_x$] and [Cu$^{II}$(OH)(H$_2$O)$_x$])[28] at ambient temperature. All spectra are characterized by two features in the high field region, one around 3130 G attributed to freely mobile isotropic Cu$^{II}$ ions, and the other around 3220 G due to Cu$^{II}$ ions with restricted mobility, which display anisotropic EPR characteristics. In the hyperfine region, anisotropic Cu$^{II}$ ions display spin Hamiltonian parameters at $g_{||}$ = 2.41, $A_{||}$ = 115 G. For the isotropic

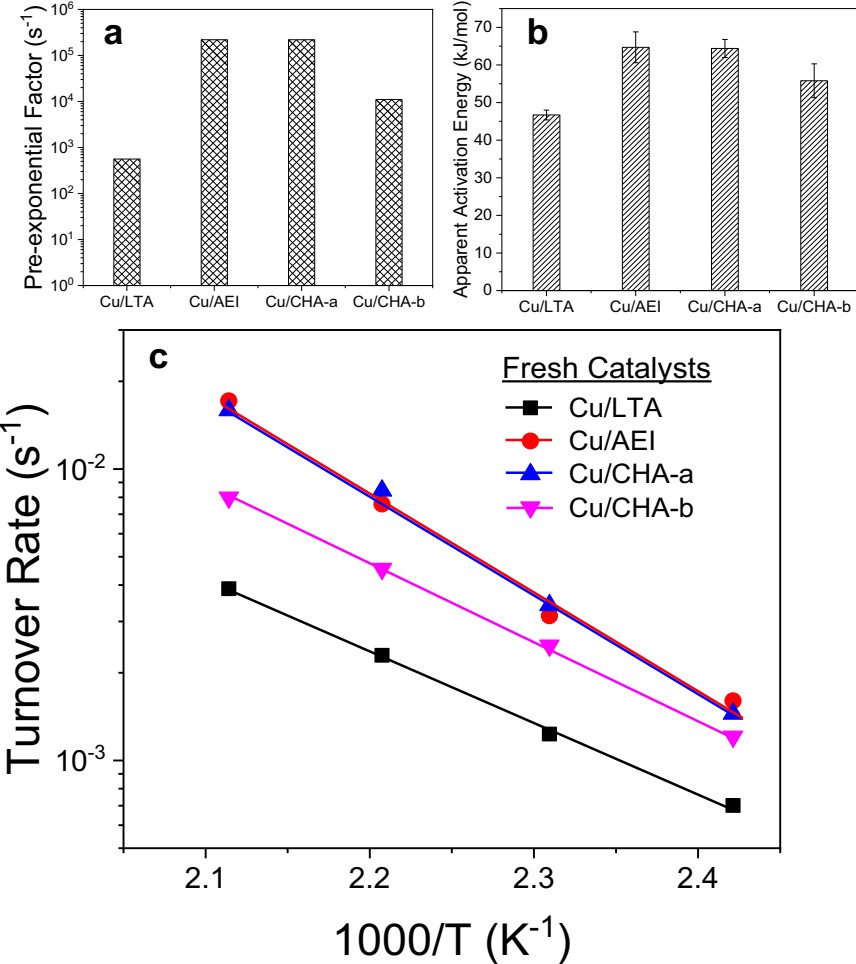

**Fig. 1 | Arrhenius analysis of the standard NH₃-SCR kinetic results. a–c** Pre-exponential factors (**a**), apparent activation energies (**b**) and turnover rates (**c**) over Cu/LTA, Cu/AEI, Cu/CHA-a, and Cu/CHA-b. Error bars in (**b**) represent standard deviation of linear regression in Arrhenius analysis. The reactant feed contains 350 ppm NOx (including ~10 ppm NO₂), 350 ppm NH₃, 2.5% H₂O, 10% O₂, and balanced N₂ at a gas hourly space velocity (GHSV) of ~2 × 10⁵ h⁻¹.

Cu$^{II}$ ions, $g$ tensors in $z$ and $x/y$ directions are inseparable, resulting in an averaged $g_{ave}$ = 2.24[18,29,30]. By $^{17}$O isotopic labeling of the zeolite framework, in conjunction with advanced EPR methodologies and DFT modeling, Bruzzese et al.[31] recently attributed the isotropic ions to [Cu$^{II}$(H$_2$O)$_6$] and anisotropic ions to [Cu$^{II}$(H$_2$O)$_4$(O$_L$)$_2$] where O$_L$ denotes lattice oxygen of the zeolite support. We note that such attributions are most certainly incomplete for the samples used here. For example, the presence of mobile [Cu$^{II}$(OH)(H$_2$O)$_x$] in our samples is clearly evidenced by H$_2$-TPR results in Supplementary Fig. 3. Each anisotropic Cu$^{II}$ species, owing to a Cu nuclear spin of $I$ = 3/2, should show well-resolved hyperfine signals of four lines with equal space and intensity. The resolution is partly lowered by the overlapping of isotropic and anisotropic signals. By comparing relative intensities of isotropic vs. anisotropic signals, it is readily concluded that the LTA support does not exert stronger restrictions than the other three supports to the mobility of solvated Cu ions; rather, the opposite appears to be the case. By cooling the samples to −150 °C to immobilize hydrated Cu ions, the spectra obtained (Fig. 2b) now only show anisotropic features, i.e., well-resolved hyperfine signals with an intensity ratio of 1:1:1:1. The high-field signals are also much narrower due to the decrease of spin-spin coupling. It is interesting to note that Cu/AEI and Cu/CHA-a/b display identical $g_{||}$ and $A_{||}$ tensor values, slightly different from those of Cu/LTA. Such small differences are induced by small variations in Cu$^{II}$ configurations in the $z$ direction, consistent with the fact that the AEI structure is more similar to CHA than to LTA.

Figure 2c shows the EPR spectra at −150 °C of the dehydrated samples by heating to 350 °C in dry N$_2$. Unlike Z$_2$Cu$^{II}$ and ZCu$^{II}$OH in hydrated state, dehydrated ZCu$^{II}$OH has been suggested to be EPR silent due to a pseudo Jahn-Teller effect[29]. More recently, Bruzzese et al. demonstrated that dehydrated ZCu$^{II}$OH is still EPR active (with $g_{||}$ = 2.29 and $A_{||}$ = 114 G), which adopts a 4 Cu-O coordination at cryogenic temperatures as opposed to a 3 Cu-O coordinated structure typically observed at higher temperatures[31]. Since such hyperfine signals are barely detected in our dehydrated samples, the loss of ZCu$^{II}$OH signals is likely due to other causes. From recent literature, ZCu$^{II}$OH has been known to lose EPR visibility during dehydration via dimerization and autoreduction chemistries[32,33]. As such, we assign all ESR signals in Fig. 2c to Z$_2$Cu$^{II}$, and tentatively suggest that the loss of EPR signal is due to ZCu$^{II}$OH conversion to EPR silent moieties. In this case, $g_{||}$ and $A_{||}$ tensor values for the Z$_2$Cu$^{II}$ in Cu/LTA (denoted α), Cu/AEI (β) and Cu/CHA-b (δ) are highly similar, due to its site anchored on 6MR with two framework Al atoms in an Al-Si-Al linkage, "*meta*"-configuration. From now on, we denote this species as $m$-Z$_2$Cu$^{II}$ (Supplementary Fig. 3). We also note that Cu/CHA-a contains another species denoted γ, i.e., Z$_2$Cu$^{II}$ next to 6MR with two framework Al ones in an Al-Si-Si-Al linkage, "*para*"-configuration ($p$-Z$_2$Cu$^{II}$). Ratios of double integrated peak areas of the spectra in Fig. 2b, c, a straightforward measure of isolated Cu$^{II}$ speciation, are presented in Fig. 2d. The high Z$_2$Cu$^{II}$ selectivity (~70%) for Cu/AEI and Cu/CHA-b is readily obtained, and it can be understood by the lower S/Al ratio (ca. 11) of their support (Supplementary Table 2). The same rationale applies to the much lower Z$_2$Cu$^{II}$

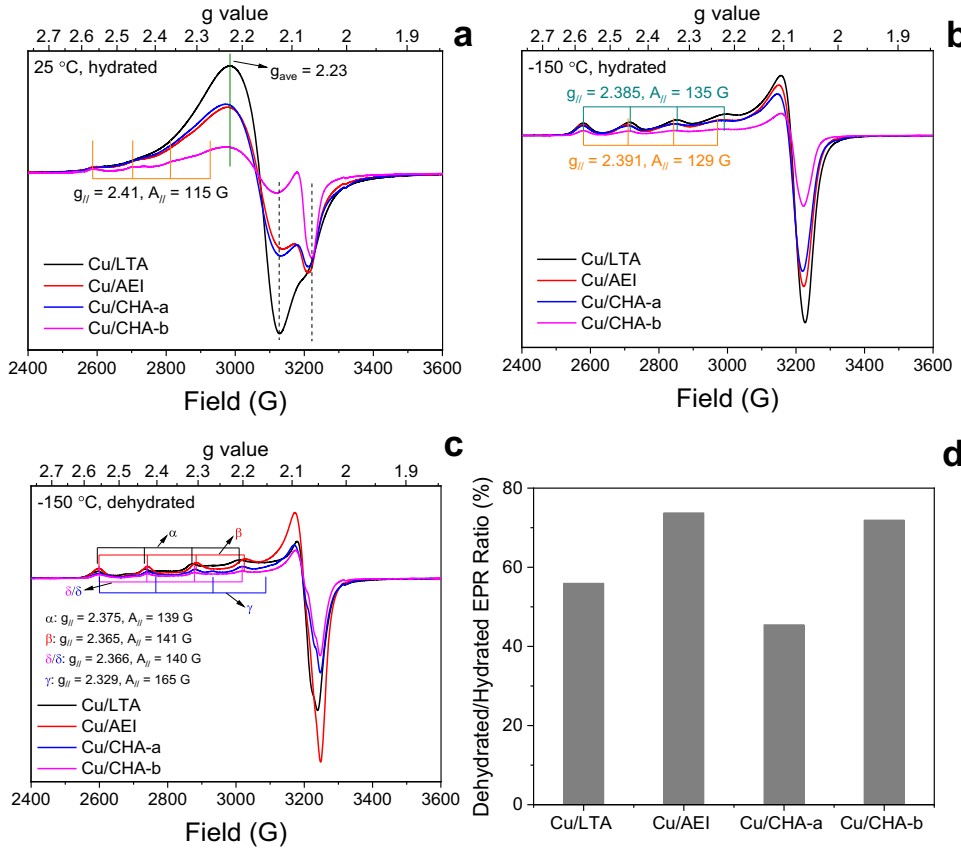

**Fig. 2 | In situ EPR spectra. a–c** Spectra of hydrated Cu-zeolites measured at 25 °C (**a**) and −150 °C (**b**) and dehydrated Cu-zeolites measured at −150 °C (**c**). **d** Double integrated peak area ratios of dehydrated and hydrated samples measured at −150 °C.

selectivity (~45%) for Cu/CHA-a. The $Z_2Cu^{II}$ selectivity (~55%) for Cu/LTA is somewhat unexpected, given the highest Al-pair density of its support (Supplementary Table 3).

***Operando* EPR spectroscopy**

Very recently, we demonstrated using operando EPR that mobility of $NH_3$-solvated $Cu^{II}$ ions in Cu/CHA catalysts serves as an indicator for their intercage transfer capacity, which closely correlates with Cu atomic efficiency during low-temperature SCR, i.e., higher Cu ion mobility corresponds to higher atomic efficiency[18]. We use the same approach here to compare the current catalysts. We note that ~2.5% cofed $H_2O$ was applied in our kinetic measurements shown above, but not during our operando EPR measurements. Even though cofed $H_2O$ has been reported to markedly influence transient SCR kinetics, its influence under steady state (applied here) is rather minimal[34,35]. As such, good correlations between our kinetics and spectroscopic studies are anticipated. The operando EPR spectra acquired over the fresh catalysts from 50 to 350 °C are presented in Supplementary Fig. 5, and those at specific temperatures (i.e., 50, 150, 225, and 350 °C) are depicted in Fig. 3. At 50 °C (below the SCR light-off temperature), Cu ions stay exclusively in a + 2 oxidation state and become solvated by $NH_3$, because of the ligand displacement from $H_2O$ to $NH_3$, as the latter binds stronger to $Cu^{II}$ ions[36]. As evidenced from Fig. 3a, not all such cations display the same mobility. Mobility restricted $Cu^{II}$ ions display anisotropic characteristics with hyperfine tensor values at $g_{||}= 2.25$, $A_{||} = 170$ G and high-field signals located around 3300 G, whereas freely mobile ions display isotropic characteristics, with $g_{ave} = 2.18$ and high-field signals around 3260 G. The mobility restricted $Cu^{II}$ ions may contain both $NH_3$ and zeolite lattice oxygen ($O_L$) ligands; the freely mobile ones are readily attributed to the combination of $Cu^{II}(NH_3)_4$ and $Cu^{II}(OH)(NH_3)_3$[13]. By comparing the relative intensities of the two

Cu species, it can be concluded that $NH_3$-solvated $Cu^{II}$ ions in Cu/LTA also display higher mobility (i.e., weaker Cu-support interactions) than their counterparts in Cu/AEI and Cu/CHA, similar to the $H_2O$-solvated $Cu^{II}$ ions in Fig. 2a. Note that at the same temperature of 50 °C, Negri et al.[37] demonstrated the formation of a mobile $[Cu^{II}(NH_3)_3(NO_3)]^+$ complex by generating $Cu^{II}$-$NO_3^-$ first, followed by $NH_3$ ligating. Under steady-state conditions applied here, the presence of this mixed-ligand species, however, is not very likely. Indeed, as Marberger et al.[38] pointed out in their XAS studies that $Cu^{II}$-$(NO_x)_y$-type species were observed, but only when the catalyst was no longer fully covered by $NH_3$ and after the disappearance of gas-phase $NH_3$. Even if small quantities of $[Cu^{II}(NH_3)_3(NO_3)]^+$ complex does form under our operando conditions at 50 °C, as a mobile complex, it will not generate any split hyperfine signals. It is only anticipated that its $g_{ave}$ signal will be completely overwhelmed by the much stronger $[Cu^{II}(NH_3)_4]$ and $[Cu^{II}(OH)(NH_3)_3]$ $g_{ave}$ signals. As such, no further discussion about this species is given below.

At 150 and 225 °C (above the SCR light-off temperature), the EPR detectable $Cu^{II}$ in Cu/AEI and Cu/CHA-a/b stay largely anisotropic (Fig. 3b, c); their limited mobility suggests that they are partly coordinated to $NH_3$ ligands, and partly to $O_L$ as $Cu^{II}(NH_3)_{4-x}(O_L)_x$[18]. However, the EPR detectable $Cu^{II}$ in Cu/LTA at these two temperatures remain largely isotropic. It is worth noting that the $g_{ave}$ value shifts from 2.18 at 50 °C to 2.27/2.28 at 150/225 °C. Since the most readily detected intermediates by operando EPR spectroscopy are arguably the least reactive ones, such a shift suggests that the highly mobile species that gives rise to a $g_{ave}$ value of 2.27/2.28 is primarily the redox resistant $Cu^{II}(NH_3)_4$. Note that $Cu^{II}(NH_3)_4$ is both theoretically suggested[13] and experimentally confirmed[18,38,39] as an intermediate in low-temperature SCR. In recent operando X-ray absorption spectroscopic (XAS) studies for SCR, low-temperature spectra were typically simulated invoking

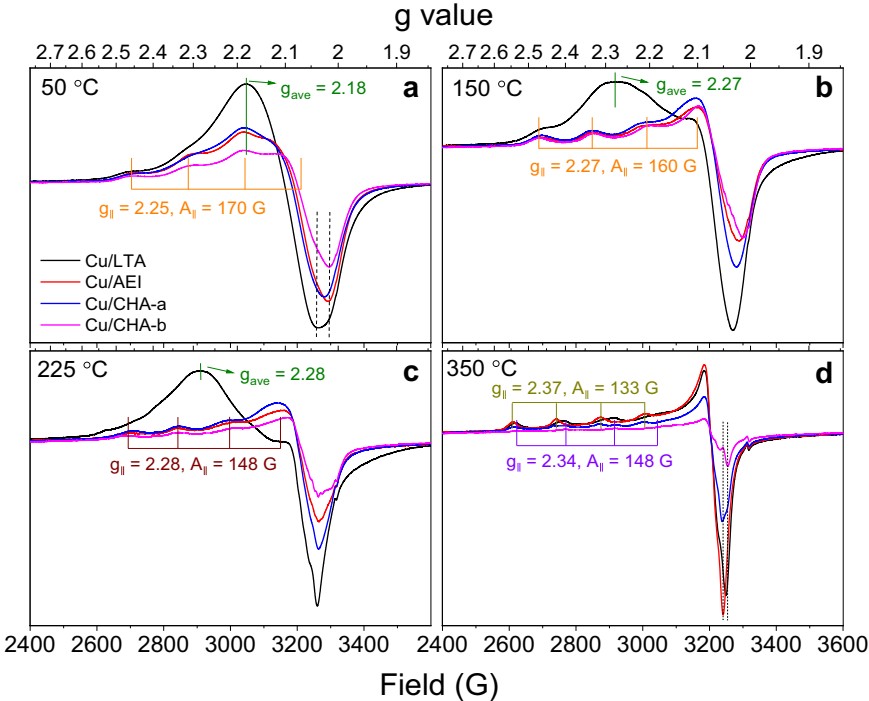

**Fig. 3 | Operando EPR spectra acquired during steady-state standard SCR over the fresh catalysts at various reaction temperatures. a–d** Spectra at 50 °C (**a**), 150 °C (**b**), 225 °C (**c**) and 350 °C (**d**). The reactant feed contains 350 ppm NOx (including ~10 ppm NO₂), 350 ppm NH₃, 10% O₂, and balanced N₂ at a gas hourly space velocity (GHSV) of ~$4 \times 10^5$ h⁻¹.

three Cu moieties, namely mobile Cu$^{II}$ (m-Cu$^{II}$, Cu$^{II}$(NH$_3$)$_4$), mobile Cu$^{I}$ (m-Cu$^{I}$, Cu$^{I}$(NH$_3$)$_2$), and zeolite-bound Cu$^{II}$ (Z-Cu$^{II}$)[38–40]. In the present study, the EPR invisibility of mobile Cu$^{II}$(NH$_3$)$_4$ on Cu/CHA and Cu/AEI at 150/225 °C, and the visibility of the same species on Cu/LTA at the same temperatures, therefore, reveal an important difference between Cu/LTA and the other three catalysts, that is, Cu$^{II}$(NH$_3$)$_4$ likely undergoes very rapid solvation-desolvation (i.e., Cu$^{II}$(NH$_3$)$_4 \rightleftharpoons x$NH$_3$ + Cu$^{II}$(NH$_3$)$_{4-x}$(O$_L$)$_x$) on CHA and AEI, rendering detectability of immobilized Cu$^{II}$(NH$_3$)$_{4-x}$(O$_L$)$_x$ alone. In contrast, this rapid interconversion does not appear to establish on LTA, rendering the detectability of primarily mobile Cu$^{II}$(NH$_3$)$_4$). In the following, DFT calculations will be used to further elucidate such a dramatic support effect. At an even higher temperature of 350 °C (Fig. 3d), all NH$_3$-ligated intermediates become sufficiently reactive, and only completely immobilized Cu$^{II}$ ions coordinated to O$_L$ are now detected. Note that the most readily detected Z$_2$Cu$^{II}$ in Cu/LTA and Cu/CHA-b are p-Z$_2$Cu$^{II}$ ($g_{||}$ = 2.34, $A_{||}$ = 148 G), whereas those in Cu/AEI and Cu/CHA-a are m-Z$_2$Cu$^{II}$ ($g_{||}$ = 2.37, $A_{||}$ = 133 G).

Contents of EPR active Cu$^{II}$ species under reaction conditions were quantified by double-integrating the obtained spectra (Supplementary Fig. 5). Plotting the ratios of higher temperature signal against that at 50 °C (which represents total Cu ion content) clearly shows that under kinetically controlled SCR conditions (e.g., 150 °C), the percentage of Cu that remains EPR visible (i.e., as Cu$^{II}$) follows the trend Cu/LTA > Cu/CHA-b > Cu/CHA-a ≥ Cu/AEI (Supplementary Fig. 6). In comparison to the Cu atomic efficiency trend shown in Fig. 1, it is readily concluded that higher EPR visibility corresponds to lower Cu atomic efficiency, i.e., the reduction half-cycle (RHC; Cu$^{II}$ → Cu$^{I}$) of the SCR redox cycle plays a stronger rate-controlling role than other possible kinetically relevant factors under such conditions. Note that in using quantitative EPR measurements to correlate Cu atomic efficiency, one limitation is that some Cu$^{II}$ moieties may become EPR silent due to, for example, to fast relaxations. As discussed above, rapid interactions with the supports render Cu$^{II}$(NH$_3$)$_4$ EPR invisible on CHA and AEI at 150/225 °C, even though the presence of Cu$^{II}$(NH$_3$)$_4$ has been

repeatedly confirmed by operando XAS studies under similar conditions[38–40]. Fortunately, the EPR visibility and Cu atomic efficiency correlation appears to hold here even with this uncertainty. Another limitation for quantitative operando EPR, as indicated by spectra shown in Fig. 3, is that spin-Hamiltonian parameters for Cu species vary rather dramatically with the nature of the ligands (N or O), and with temperature. As such, detailed quantitative description of Cu species via spectrum simulation using linear combination fit of model species as in the case of operando XAS, is not yet achievable. However, spin-Hamiltonian parameters of SCR relevant model species, e.g., Cu$^{II}$(NH$_3$)$_4$, Cu$^{II}$(OH)(NH$_3$)$_3$, or even the [Cu$^{II}$(NH$_3$)$_3$(NO$_3$)]$^+$ complex first prepared by Negri et al.[37], can be readily measured at cryogenic temperatures. In this case, EPR spectra acquired on working SCR catalysts rapidly quenched to the same temperatures may be simulated using spin-Hamiltonian parameters of such model Cu species.

## DFT calculations

According to the recent studies by Tronconi and co-workers, RHC is facilitated not only by the facile hydrolysis of redox resistant Cu$^{II}$(NH$_3$)$_4$ to redox active Cu$^{II}$(OH)(NH$_3$)$_3$[22], but also by the cohabitation of two Cu$^{II}$(OH)(NH$_3$)$_3$ within the same *cha* cage[17]. To investigate differences in the Cu$^{II}$(NH$_3$)$_4$ hydrolysis and Cu$^{II}$(OH)(NH$_3$)$_3$ migration within Cu/CHA and Cu/LTA, we performed DFT calculations. As suggested previously[13], the most prevalent Cu$^{II}$ species under typical low-temperature NH$_3$-SCR conditions is Cu$^{II}$(NH$_3$)$_4$, which hydrolyzes to more reactive Cu$^{II}$(OH)(NH$_3$)$_3$ in the presence of H$_2$O, via Cu$^{II}$(NH$_3$)$_4$ + H$_2$O → Cu$^{II}$(OH)(NH$_3$)$_3$ + NH$_4^+$. Our proposed hydrolysis pathway contains the following key steps: (i) the displacement of an NH$_3$ ligand in Cu$^{II}$(NH$_3$)$_4$ with H$_2$O to form Cu$^{II}$(H$_2$O)(NH$_3$)$_3$; (ii) the transition state formation between Cu$^{II}$(H$_2$O)(NH$_3$)$_3$ and the detached NH$_3$ molecule via H-O⋯H⋯NH$_3$; and (iii) the formation of Cu$^{II}$(OH)(NH$_3$)$_3$ + NH$_4^+$. As shown in Fig. 4a, the hydrolysis process within the cylindrical *cha* cage is exergonic while it is endergonic in the considerably larger but spherical *lta* cage. Furthermore, the calculated Gibbs free energy of activation in Cu/CHA is 41.8 kJ mol⁻¹, lower than

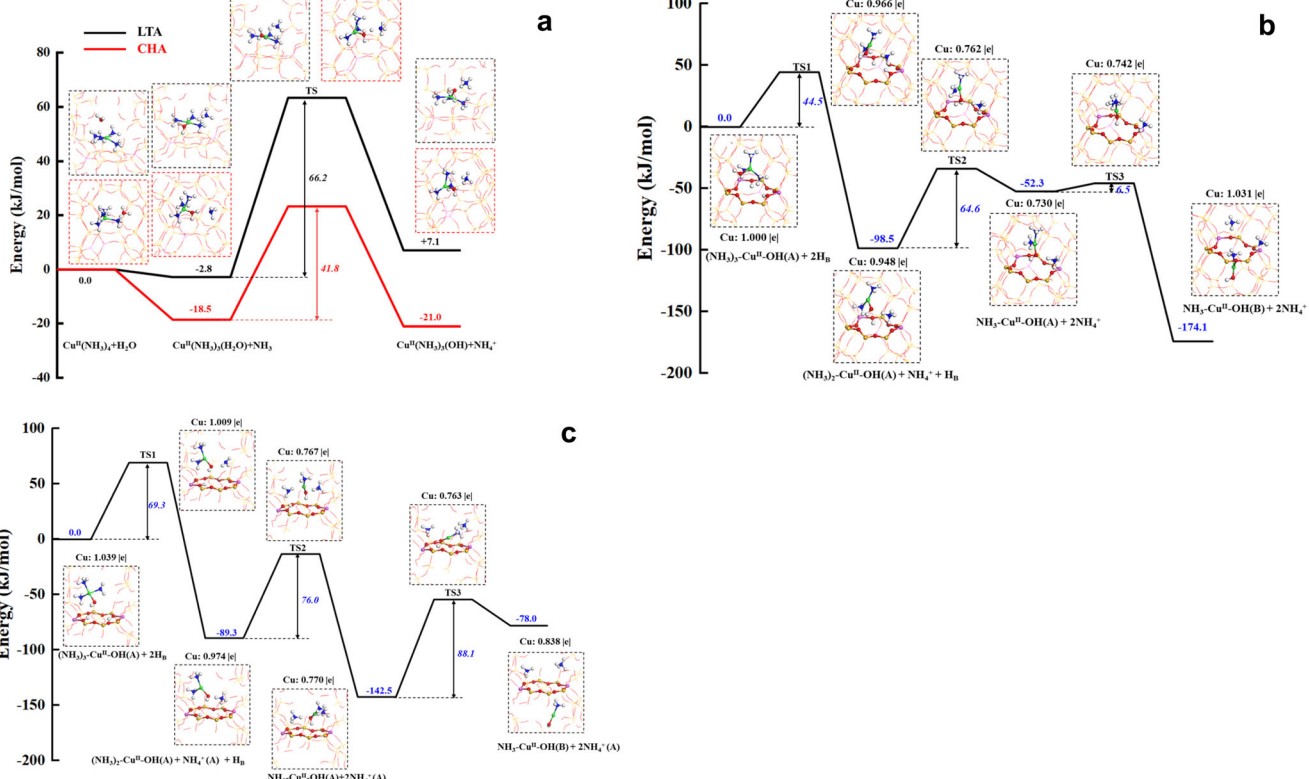

**Fig. 4 | DFT calculated energetics along reaction coordinations. a** comparison of hydrolysis processes of $Cu^{II}(NH_3)_4$ to $Cu^{II}(OH)(NH_3)_3$ in LTA and CHA zeolites. **b** diffusion of $Cu^{II}(OH)(NH_3)_3$ intermediate through CHA 8MR via de-solvation path

$Cu^{II}(OH)(NH_3)_3$ (A) $+ 2H_B^+ \rightarrow Cu^{II}(OH_2)(NH_3)_3$ (A) $+ H_B^+ \rightarrow Cu^{II}(NH_3)_2$ (A) $+ H_2O + NH_4^+ \rightarrow Cu^{II}(NH_3)(B) + H_2O + NH_4^+$. **c** the same diffusion process via LTA 8MR.

that of 66.2 kJ mol$^{-1}$ in Cu/LTA. As such, $Cu^{II}(NH_3)_4$ hydrolysis to $Cu^{II}(H_2O)(NH_3)_3$ is much more facile in Cu/CHA than in Cu/LTA. Our calculations may be far from being exhaustive since there exist multiple framework Al configurations to stabilize $Cu^{II}(NH_3)_4$ in the periodic Cu/CHA and Cu/LTA models with two unit cells. Nevertheless, the calculation results are in remarkable agreement with the operando EPR data in Fig. 3 where, at 150 and 225 °C, isotropic $Cu^{II}(NH_3)_4$ is only observed in Cu/LTA. This can be attributed to the lower BAS strength of the LTA support (Supplementary Fig. 4).

As described above, RHC speeds up when two $Cu^{II}(OH)(NH_3)_3$ cohabitate within the same *cha* cage[17]. This led us to simulate the cage-to-cage migration of $Cu^{II}(OH)(NH_3)_3$ through the 8MRs of Cu/CHA and Cu/LTA as an essential step for such a cohabitation to exist. Consistent with previous simulations, our simulation results reveal that $Cu^{II}(OH)(NH_3)_3$ itself is too bulky to diffuse through 8MRs[13,17]. Therefore, it is most likely that $Cu^{II}(OH)(NH_3)_3$ first de-solvates to $Cu^{II}(OH)(NH_3)_3$, and then diffuses through 8MRs that contains two BAS sites, i.e., $Cu^{II}(OH)(NH_3)_3$ (A) $+ 2H_B^+ \rightarrow Cu^{II}(OH)(NH_3)_2$ (A) $+ NH_4^+ + H_B^+ \rightarrow Cu^{II}(OH)(NH_3)$ (A) $+ 2NH_4^+ \rightarrow Cu^{II}(OH)(NH_3)$ (B) $+ 2NH_4^+$. As shown in Fig. 4b, this process is exergonic in Cu/CHA, with an energy gain of 174.1 kJ mol$^{-1}$. Calculated Gibbs free energies of activation for the three transition states are 44.5, 64.6 and 6.5 kJ mol$^{-1}$, respectively. The diffusion through 8MRs in Cu/LTA is also exergonic, but with a lower energy gain of 78.0 kJ mol$^{-1}$ (Fig. 4c). Activation barriers to overcome the three transition states are 69.3, 76.0 and 88.1 kJ mol$^{-1}$, respectively. These energy differences suggest the kinetically more feasible nature of diffusion through 8MRs in Cu/CHA. Bader charges of the Cu centers in Fig. 4b and c, normalized against that of $Cu^{II}(OH)(NH_3)$ in CHA as 1.000|e|, also demonstrate that during the diffusion process, Cu maintains a + 2 oxidation state, i.e., diffusion is not redox driven. An alternative de-solvation and diffusion pathway is $Cu^{II}(OH)(NH_3)_3$ (A) $+ 2H_B^+ \rightarrow Cu^{II}(OH_2)(NH_3)_3$ (A) $+ H_B^+ \rightarrow Cu^{II}(NH_3)_2$ (A) $+ H_2O + NH_4^+ \rightarrow$

$Cu^{II}(NH_3)_2(B) + H_2O + NH_4^+$. For both Cu/CHA and Cu/LTA, this latter path was found energetically more demanding and thus less likely (Supplementary Fig. 7).

It is important to note from the calculations above that the energy barriers for the hydrolysis and intercage diffusion of the associated $Cu^{II}$-amine complexes are around the same magnitude as standard SCR activation energies (typically 60–80 kJ/mol). As such, the hydrolysis and intercage diffusion processes can certainly play kinetically relevant rules, i.e., the redox chemistry does not always play sole rate-limiting roles during low-temperature standard SCR. The low-temperature SCR kinetics, operando EPR and DFT studies presented above clearly show the kinetic relevance of zeolite BAS strength, i.e., at low BAS strength, $Cu^{II}(NH_3)_4$ hydrolysis to $Cu^{II}(OH)(NH_3)_3$ is no longer so facile as not to display any kinetic consequences. Previous theoretical studies also predict that BASs catalyze reaction intermediates (e.g., H$_2$NNO) spilled over from Cu[15,19]. However, such reactions appear to have exceedingly low activation energies, largely ruling out their rate-limiting roles. It has been repeatedly shown that hydrothermal aging under harsh conditions largely eliminates the residual zeolite BASs via dealumination[41–43]. This implies that the kinetic influence from BAS will be smaller for hydrothermally aged catalysts, lowering influences of BAS strength difference to Cu atomic efficiency. These notions are demonstrated below.

## Hydrothermal aging effects

To obtain additional insights into the kinetic role of zeolite BAS strength after hydrothermal aging, we carried out a SCR kinetics and operando EPR study of hydrothermally aged catalysts. Due to high hydrothermal stability of the catalysts employed here, we selected an aging temperature of 850 °C, harsh enough to extensively dealuminate the supports, but not too harsh to destroy their overall structural integrity. Comparison of the isolated $Cu^{II}$ ion contents of the

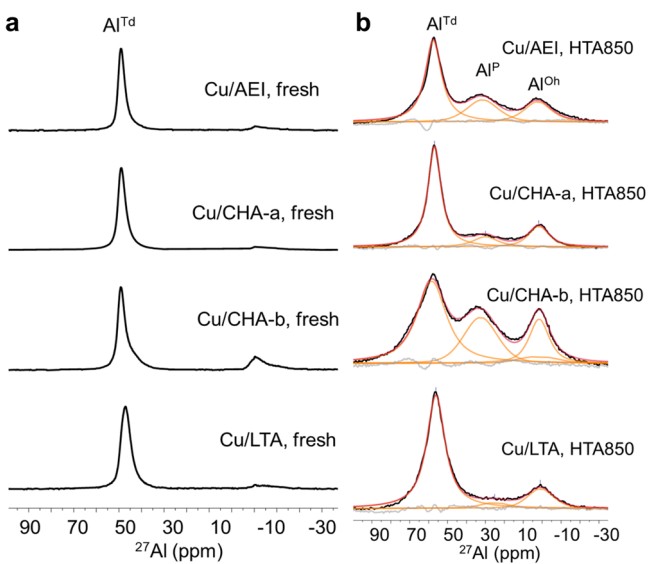

**Fig. 5 | $^{27}$Al MAS NMR spectra. a, b** Spectra of fresh (**a**) and hydrothermally aged (**b**) Cu-zeolites.

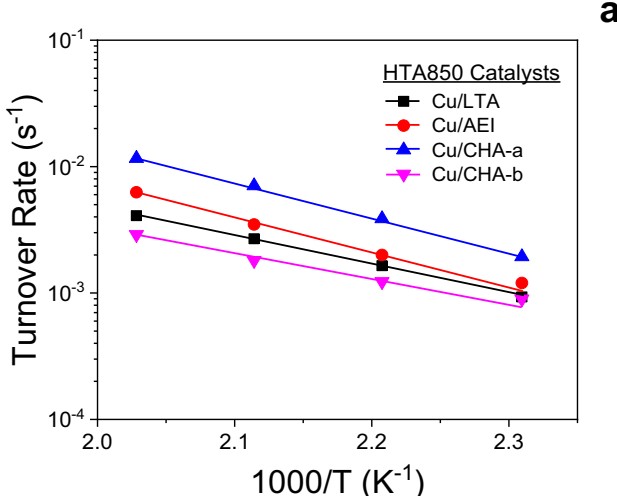

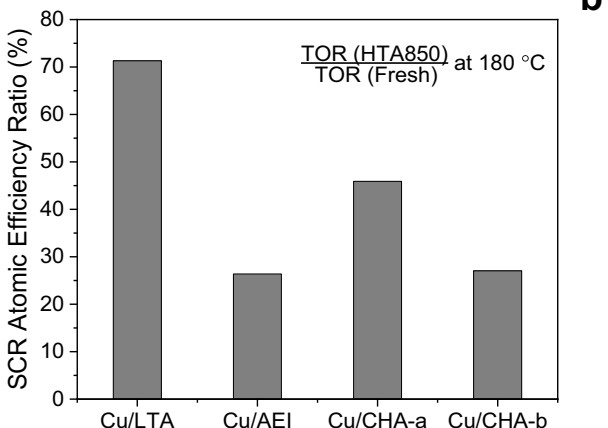

**Fig. 6 | Standard NH$_3$-SCR kinetic results of the hydrothermally aged catalysts.**
**a, b** Turnover rates (TORs) in the form of Arrhenius plots (**a**) and turnover rate (TOR) ratios at 180 °C between fresh and hydrothermally aged catalysts (**b**).

hydrothermally aged catalysts, labeled "HTA850", with those of the fresh catalysts clearly shows that aging causes partial loss of isolated Cu$^{II}$ ions via formation of EPR silent Cu moieties, e.g., Cu$_x$O$_y$ clusters and Cu-aluminates[44] (Supplementary Tables 2 and 4). The highest isolated Cu$^{II}$ ion loss is found on Cu/LTA, followed by Cu/CHA-b, Cu/CHA-a and Cu/AEI, partly reflected by the H$_2$-TPR results in Supplementary Fig. 8. Note also that NH$_3$ desorption from BASs greatly decreases after hydrothermal aging, a strong indication for support dealumination (Supplementary Figs. 4 and 9). When integrating the NH$_3$-TPD peak areas above 350 °C (which can be roughly regarded as desorption from BASs) for fresh and aged Cu/CHA-a, Cu/CHA-b and Cu/AEI catalysts, their dealumination levels were estimated to be about 75%, 85% and 80%, respectively. For Cu/LTA, since NH$_3$ desorption from Cu and BASs overlaps intimately, the same estimation proves difficult.

Figure 5 compares the $^{27}$Al magic angle spinning (MAS) NMR spectra for the fresh and HTA850 catalysts. The fresh catalysts display a prominent $^{27}$Al signal around 50 ppm due to framework tetrahedral Al (Al$^{Td}$) atoms, together with a much weaker signal around 0 ppm corresponding to extraframework octahedral Al (Al$^{Oh}$) species[27,42]. However, the spectra of all HTA850 catalysts are characterized by an additional, broad $^{27}$Al signal around 30 ppm, assignable to penta-coordinated Al (Al$^P$) species[45,46]. Comparison of the relatively intensities of the Al$^{Td}$, Al$^P$ and Al$^{Oh}$ signals (Supplementary Table 5) shows that hydrothermal stability of the catalysts follows the order Cu/LTA > Cu/CHA-a > Cu/AEI > Cu/CHA-b, as previously reported[6,7]. This suggests its dependence on both the framework topology and composition of zeolite supports.

The formation of side products NO$_2$ and N$_2$O during steady-state standard NH$_3$-SCR over the HTA850 catalysts can be found in Supplementary Fig. 10, and their TORs, obtained from the first-order kinetic analysis of light-off SCR data, are presented in Fig. 6a. The atomic efficiency for Cu sites in aged Cu/LTA is now much closer to the efficiencies for aged Cu/CHA and Cu/AEI. This confirms our claim that zeolite BASs play important rate-controlling roles; by eliminating such sites via dealumination, their kinetic influence is lowered. The TOR values between the fresh and HTA850 catalysts at 180 °C in Fig. 6b reveal that the aged catalysts display remaining atomic efficiency order Cu/LTA > Cu/CHA-a > Cu/AEI > Cu/CHA-b, which further supports the kinetic relevance of BASs. It should be noted here that the Cu$^I$ ions located within sterically inaccessible *sod* cages of Cu/LTA are migrated to the unoccupied 6-membered rings (6MRs) in much larger and thus readily accessible *lta* cages, while accompanying their oxidation to Cu$^{II}$ ions during hydrothermal aging at 750–900 °C, which has been responsible for the unusual increase in low-temperature NH$_3$-SCR activity[9] and thus in Cu atomic efficiency. As such, the zeolite cage architecture would be another crucial factor in explaining the highest remaining Cu atomic efficiency for aged Cu/LTA.

The DFT results in Fig. 4, as well as prior theoretical studies[17,22], strongly suggest that zeolite BASs influence low-temperature SCR rates by their participation in Cu$^{II}$(NH$_3$)$_4$ hydrolysis and intercage transfer of Cu$^{II}$(OH)(NH$_3$)$_3$. To confirm this, we performed operando EPR measurements on the HTA850 catalysts at different temperatures (Supplementary Fig. 11). The operando EPR spectra acquired at 50, 150, 225, and 350 °C are compared in Fig. 7. All spectra at 50 °C (Fig. 7a) display a marked similarity in lineshape; the hyperfine region contains predominantly isotropic features at $g_{ave}$ = 2.18 due to highly mobile Cu$^{II}$(NH$_3$)$_4$ and Cu$^{II}$(OH)(NH$_3$)$_3$ species, and very weak anisotropic features at $g_{\parallel}$ = 2.25, $A_{\parallel}$ = 170 G due to NH$_3$-solvated Cu$^{II}$ ions with constrained mobility. As compared to the spectra measured at the same temperature for the fresh catalysts (Fig. 3a), it is evident that zeolite support dealumination weakens the interactions between the zeolite framework and Cu$^{II}$ ions, rendering them more mobile. Even at 150 °C, the aged catalysts are still characterized by largely isotropic signals, where both Cu/CHA and Cu/AEI display a tensor value at

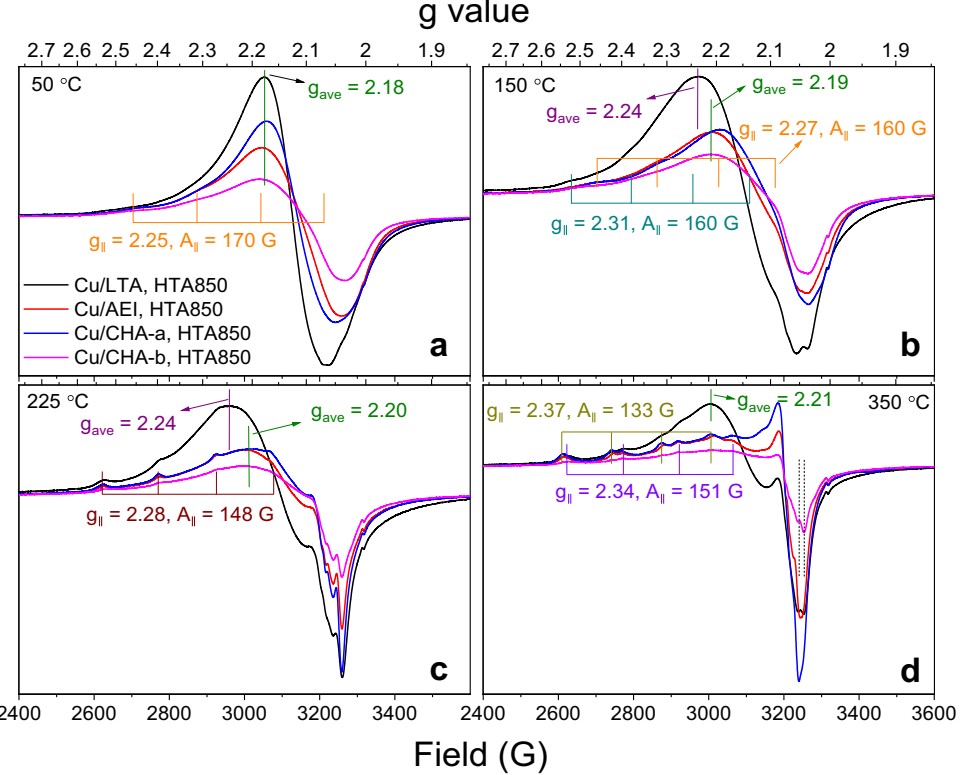

**Fig. 7 | *Operando* EPR spectra acquired during steady-state standard SCR over the HTA850 catalysts at various reaction temperatures. a–d** Spectra at 50 °C (**a**), 150 °C (**b**), 225 °C (**c**) and 350 °C (**d**). Reactant concentrations in the feed are the same as those in Fig. 3.

$g_{ave} = 2.19$ and Cu/LTA with $g_{ave} = 2.24$ (Fig. 7b). This difference indicates that Cu/CHA and Cu/AEI contain higher concentrations of mobile $Cu^{II}(OH)(NH_3)_3$ species than Cu/LTA, suggesting that the residual BASs in Cu/CHA and Cu/AEI are still more active in promoting $Cu^{II}(NH_3)_4$ hydrolysis than the corresponding sites in Cu/LTA. The aged catalysts also show weak anisotropic signals with tensor values $g_{||} = 2.27$, $A_{||} = 160$ G and $g_{||} = 2.31$, $A_{||} = 160$ G, respectively (Fig. 7b), due to $NH_3$-solvated $Cu^{II}$ ions with constrained mobility, which is opposite to the operando EPR results in Fig. 3b that only $Cu^{II}$ ions with constrained mobility are detectable on fresh Cu/CHA and Cu/AEI at 150 °C. It thus appears that mobile $Cu^{II}(OH)(NH_3)_3$ species in the aged catalysts become less reactive than the same species in the fresh ones. According to a recent proposal by Gao and Tronconi and their co-workers, low-temperature SCR rate is facilitated by pairs of $Cu^{II}(OH)(NH_3)_3$ species cohabitated in the same zeolite cage[17]. As such, the reactivity decrease of $Cu^{II}(OH)(NH_3)_3$ in aged catalysts can be correlated with their decreased intercage transfer capability. In other words, $Cu^{II}(OH)(NH_3)_3$ cannot migrate through 8MRs without partial desolvation that is facilitated by the framework Al atoms on 8MRs (Fig. 4b, c).

The operando EPR spectra measured at 225 °C still display combined isotropic and anisotropic signals (Fig. 7c). As compared to spectra measured at 150 °C, intensities of the hyperfine region anisotropic signals increase for all catalysts, and the high-field region signals also appear to be better resolved. All such changes suggest a decrease in overall Cu mobility at 225 °C as the extent of $NH_3$ solvation decreases with increasing temperature[13]. We note that Cu/LTA still contains much higher concentrations of mobile $Cu^{II}(NH_3)_4$ than the other three samples, as evidenced by the appearance of strong signals at $g_{ave} = 2.24$ in the hyperfine region. At 350 °C where aged Cu/CHA and Cu/AEI only display completely de-solvated $Cu^{II}$ species as p-$Z_2Cu^{II}$ ($g_{||} = 2.34$, $A_{||} = 151$ G) and m-$Z_2Cu^{II}$ ($g_{||} = 2.37$, $A_{||} = 133$ G), Cu/LTA still displays strong isotropic signals at $g_{ave} = 2.21$ (Fig. 7d). Compared with

the signals at $g_{ave} = 2.24$ observed at 225 °C, their shift to $g_{ave} = 2.21$ indicates that $Cu^{II}(NH_3)_4$ hydrolysis to $Cu^{II}(OH)(NH_3)_3$ finally becomes feasible in Cu/LTA at this high temperature. This finding is in line with the DFT results in Fig. 4a that the lower BAS strength for LTA renders $Cu^{II}(NH_3)_4$ hydrolysis much more difficult than that in CHA and AEI.

The ratios of EPR-visible $Cu^{II}$ and total Cu at various reaction temperatures during steady-state SCR over the HTA850 catalysts are displayed in Supplementary Fig. 12. Under kinetically controlled SCR conditions (e.g., 150 °C), the percentage of Cu, which remains EPR visible (i.e., as $Cu^{II}$), follows the trend Cu/LTA > Cu/CHA-b > Cu/AEI > Cu/CHA-a. Therefore, compared with the low-temperature SCR kinetics data in Fig. 6, it is evident that the HTA850 catalysts holds higher EPR visibility and thus lower Cu atomic efficiency than the corresponding fresh catalysts at any temperature. This is consistent with the above discussions that $Cu^{II}(NH_3)_4$ and $Cu^{II}(OH)(NH_3)_3$ sites in HTA850 samples are less reactive than their counterparts in the fresh catalysts, largely due to the decreased engagement from BASs, as well as to the isolated Cu ion content decrease by hydrothermal aging.

Finally, regarding the interplay between RHC and OHC in low-temperature SCR, recent transient kinetic studies by researchers from Milano[17,34] and Cummins[47] made important discoveries by isolating RHC and OHC with $NH_3 + NO$ and $NO + O_2$ titrations, respectively, and then combining the two to generate the overall redox kinetic model. In these studies, the authors used catalysts from major industrial SCR catalyst suppliers (BASF, Johnson-Matthey Inc.). We note that such catalysts are performance optimized; their redox active $ZCu^{II}OH$ contents are typically high, and redox resistant $Z_2Cu^{II}$ contents are typically low. Furthermore, $Z_2Cu^{II}$ hydrolysis to $ZCu^{II}OH$ under SCR conditions appears to be facile for these catalysts[22]. As such, catalytic function of all Cu-ions can be considered equal (e.g., all $Cu^{II}$-ions are readily reduced during the RHC step[34]), and a mean-field kinetic model works well for describing OHC[47]. In addition to transient kinetic studies, the interplay between RHC and OHC can also be systematically

probed under steady-state conditions using catalysts with a wide range of active Cu densities, and/or $O_2$ partial pressures[12,14,18,48], where OHC rate-controlling can be purposely designed, e.g., by dramatically lowering Cu concentration[18], or by lowering $O_2$ pressure during SCR reaction[48]. In some of such studies, non-mean-field behavior was found to better describe OHC[14]. More generally, these latter studies demonstrate that the mechanisms of RHC and OHC do not appear to change with Cu density; rather the relative intrinsic rates of these processes change with reaction conditions, and neither one solely limits the rates of the SCR redox cycle[18,48]. This notion is again corroborated by our operando EPR quantification data in Supplementary Fig. 6, showing that neither $Cu^{II}$ nor $Cu^{I}$ is in absolute dominance under low-temperature SCR conditions, suggesting that neither RHC nor OHC is clearly the rate limiting step. In the present study, we show that support BAS strength and density play important roles on RHC rates. These new findings broaden our understanding on RHC rate-controlling by incorporating support topology and hydrothermal aging effects. Because of the EPR silent nature of $Cu^{I}$-ions and dimeric Cu intermediates that are involved in OHC, operando EPR cannot be applied to address this half-cycle. In a very recent publication by Gounder and coworkers[49], the authors demonstrated that increasing the zeolite support Al density leads to systematic increases in both the fraction of $Cu^{I}$ ions that are SCR active and OHC rate constants. Based on this new discovery and the current study, BAS influences both the fractions of SCR active $Cu^{II}$ during RHC and the fractions of SCR active $Cu^{I}$ during OHC. It is anticipated that catalyst composition and the history of use/treatment determine which effect plays more important kinetic roles under a given SCR reaction condition, and this will be further addressed in our future studies.

To summarize, low-temperature standard $NH_3$-SCR over Cu-exchanged zeolites occurs on $NH_3$-solvated Cu ions within zeolite cages via redox mechanisms in a quasi-homogeneous manner. Below the SCR light-off temperature, Cu ions stay exclusively as $NH_3$-solvated $Cu^{II}$ due to insurmountable activation barriers of the reduction half-cycle (RHC; $Cu^{II} \rightarrow Cu^{I}$) of their redox chemistry. This indicates that the oxidation half-cycle (OHC; $Cu^{I} \rightarrow Cu^{II}$) only influences SCR rates via kinetic, but not thermodynamic, control. A combined SCR kinetics and operando EPR study above the SCR light-off temperature demonstrates that the visibility of Cu by EPR correlates inversely with atomic efficiency of Cu in SCR, which can be rationalized by invoking two mechanisms reported in the recent literature that can delay RHC: (i) hydrolysis of redox resistant $Cu^{II}(NH_3)_4$ to redox active $Cu^{II}(OH)(NH_3)_3$; and (ii) intercage transfer of $Cu^{II}(OH)(NH_3)_3$. The DFT results show that both processes are energetically more feasible in Cu/CHA than in Cu/LTA, implying the kinetic relevance of BASs. The latter notion has been further corroborated by the kinetics and operando EPR data of hydrothermally aged catalysts with minimal residual BASs. A lower BAS strength of the LTA support, with one large *lta* and one much smaller *sod* cages, has similar negative kinetic effects: a decrease in $Cu^{II}(NH_3)_4$ hydrolysis and subsequently in intercage transfer of $Cu^{II}(OH)(NH_3)_3$. Hydrothermal stability of the catalysts, on the other hand, is influenced by a stronger extent from support composition and topology. To our knowledge, this study is the first to provide molecular-level insights into the SCR rate-controlling step from Cu redox, Cu transfer and support acidity, where the support acidity facilitates the formation and intercage transfer of Cu intermediates that are most relevant to Cu redox.

## Methods
### Catalyst synthesis
Two SSZ-13 zeolites (CHA-a and CHA-b) with Si/Al ~ 17 and 11, respectively, were synthesized using N,N,N-trimethyl-1-adamantyl ammonium hydroxide (TMAdaOH; 25%, Sachem) as a structure directing agent (SDA), Al(OH)$_3$ (~54% Al$_2$O$_3$; Sigma Aldrich) as an Al source, LUDOX AS-30 colloidal silica (30 wt% suspension in H$_2$O, Sigma

Aldrich) as a Si source and NaOH (≥99%, Sigma Aldrich) for adjusting pH, and d.i. water. The molar composition of the synthesis gel was 1.0 TMAdaOH:1.0 NaOH:$x$ Al$_2$O$_3$:10 SiO$_2$:220 H$_2$O, where $x$ varies to allow different Si/Al ratios. More details of the synthesis procedure can be found elsewhere[44]. An SSZ-39 (AEI) with Si/Al ~ 10 was synthesized using N,N´-dimethyl-3.5-dimethylpiperidinium hydroxide (DMPOH, Sachem) as a SDA and FAU zeolite with Si/Al~15 (CBV720, Zeolyst) as Si and Al sources, following a procedure previously reported[42]. The molar composition of the synthesis gel was 1.0 DMPOH:1.0 NaOH:5.0 SiO$_2$:0.165 Al$_2$O$_3$:46 H$_2$O. An LTA zeolite with Si/Al ~ 16 was synthesized using 1,2-dimethyl-3-(4-methylbenzyl)-imidazolium hydroxide (DMIOH) and tetramethylammonium hydroxide (TMAOH) as co-SDAs, with the synthesis mixture molar composition 0.5 DMIOH:0.067 TMAOH:0.50 HF:1.0 SiO$_2$:0.034 Al$_2$O$_3$:5.0 H$_2$O, at 175 °C for 17 h[8]. The synthesis also included the use of small amount (4 wt% of the silica in the synthesis mixture) of calcined LTA with Si/Al = 23 as seed crystals. As-synthesized zeolites were calcined in static air at 540 or 650 °C for 5 h to burn the SDAs, converted to their NH$_4$-form by thorough exchange with 0.1 M NH$_4$NO$_3$ (98%, Sigma Aldrich) solutions at 80 °C, and dried in flowing N$_2$ at 80 °C before further use.

Cu/CHA-a/b, Cu/AEI and Cu/LTA catalysts were synthesized via solution ion exchange. In their preparation, 5.0 g of NH$_4$-form zeolite was first dispersed in 100 mL of deionized water under stirring at 400 rpm. A designated amount of Cu(NO$_3$)$_2$·2.5 H$_2$O (99.9%, Sigma Aldrich) was then added to the suspension. Following which, the pH of the suspension was adjusted to ~2.5 with 0.1 M HNO$_3$ (ACS reagent grade, Sigma Aldrich) solutions. Ion exchange was carried out at 80 °C for 5 h under stirring. The solid was then recovered by centrifugation and thoroughly washed with d.i. water. After that, the solid was dried under a N$_2$ flow at 80 °C overnight and calcined in static air at 650 °C for 5 h before further use. The Si, Al and Cu contents of the catalysts were determined by Inductively Coupled Plasma Atomic Emission Spectroscopy (ICP-AES) at Galbraith Laboratories (Knoxville, TN, USA).

Hydrothermal aging of the catalysts was carried out in flowing air containing 10% water vapor at 850 °C for 24 h. The catalysts thus treated are denoted "HTA850".

### Steady-state SCR test
Standard $NH_3$-SCR ($4NO + 4NH_3 + O_2 = 4 N_2 + 6H_2O$) reaction was carried out in a plug-flow reactor system described earlier[11,50]. 120 mg of sieved catalyst (60–80 mesh) was used for the reaction tests. The composition of the gas feed included 350 ppm $NH_3$, 350 ppm $NOx$ (containing ~10 ppm $NO_2$), 10% $O_2$, 2.5% $H_2O$, and balance $N_2$. The total flow rate was 600 mL min$^{-1}$ and the gas hourly space velocity (GHSV) was estimated to be ~2 × 10$^5$ h$^{-1}$. Concentrations of reactants and products were measured by an MKS MultiGas 2030 FTIR gas analyzer with the gas cell retained at 191 °C. Steady-state measurements were conducted between 550 and 100 °C. At each target temperature, the reaction was maintained for at least 1 h to reach steady state. The following equations were used to calculate $NOx$ and $NH_3$ conversions:

$$NO_x\,Conversion\% = \frac{(NO + NO_2)_{inlet} - (NO + NO_2 + N_2O)_{outlet}}{(NO + NO_2)_{inlet}} \times 100 \tag{1}$$

$$NH_3\,Conversion\% = \frac{(NH_3)_{inlet} - (NH_3)_{outlet}}{(NH_3)_{inlet}} \times 100 \tag{2}$$

Low-temperature $NOx$ conversion data were further treated using a first-order kinetic equation $r = \frac{F}{W}(-\ln(1 - x))$, where $F$ is the $NOx$ flow rate (moles of $NOx$ per s), $W$ the mass of the catalyst (g), and $x$ the $NOx$ conversion. Arrhenius equation, $k = \frac{r}{[NOx]_0} = Ae^{\frac{-E_a}{RT}}$, was used for calculating rate constants $k$, pre-exponential factor ($A$) and apparent activation energy ($E_a$), where $[NOx]_0$ is the molar concentration of $NOx$ in

the feed[11,51,52]. Turnover rates (TORs) were also calculated using the first-order kinetic equation above, however, normalized to the molar content of isolated Cu ions rather than the mass of the catalyst.

## Catalyst characterization

Surface areas (BET method) and micropore volumes (t-plot method) of the catalysts were measured on a Quantachrome Autosorb-6 analyzer with liquid $N_2$ adsorption. Prior to analysis, the catalysts were outgassed under high vacuum overnight at 250 °C. Powder X-ray diffraction (PXRD) measurements were carried out on a Philips PW3040/00 X'Pert diffractometer with Cu Kα radiation (λ = 1.5406 Å). Data were collected with 2θ ranging from 5° to 50° using a step size of 0.01°.

Cu contents of the catalysts were measured by ICP-AES as described above. The nature of Cu in these catalysts was probed by two methods. Isolated $Cu^{II}$ ion contents were measured by electron paramagnetic resonance (EPR) on a Bruker E580 X-band spectrometer. Based on previous studies, both $Z_2Cu^{II}$ and $ZCu^{II}OH$, when fully hydrated, are EPR active and can be readily quantified[50,53]. Typically, ~10 mg of an ambient hydrated catalyst was loaded into the quartz EPR tube, and continuous scans of the sample were performed at −150 °C. The acquired spectra were double-integrated to obtain signal areas, which are proportional to the EPR-active isolated $Cu^{II}$ ion content. To quantify these, a series of standard solutions with different isolated $Cu^{II}$ ion concentrations were prepared by dissolving $Cu(NO_3)_2 \cdot 2.5H_2O$ and imidazole (Sigma Aldrich, 99.0%) in ethylene glycol (Sigma Aldrich, 99.8%). The linear calibration curve generated using the integrated area and Cu content of the standard solutions was then used for the quantification of isolated $Cu^{II}$ ions in the catalysts[44]. Good accuracy for EPR quantification (typical uncertainties within 5%) has been verified by Cu-ion quantification via other techniques, e.g., $H_2$ temperature-programmed reduction ($H_2$-TPR)[54].

To further probe the nature of Cu, the catalysts were also subjected to $H_2$-TPR experiments, carried out on a Micromeritics AutoChem 2920 apparatus. Ambient samples (~100 mg) were used for these measurements; a dehydration pretreatment was not performed to avoid autoreduction or dehydration condensation of $ZCu^{II}OH$ sites[44,55]. $H_2$-TPR was conducted in 5% $H_2$/Ar at a flow rate of 30 mL min⁻¹. The samples were ramped from ambient to 1200 °C at a ramping rate of 10 °C min⁻¹, and maintained at 1200 °C for 30 min for complete reduction. $H_2$ consumption was measured by a thermally conductive detector. CuO (99.9%, Sigma Aldrich) was used for quantification. $NH_3$-TPD measurements were carried out on a Micromeritics AutoChem 2920 apparatus equipped with an MKS Cirrus 2 quadrupole mass spectrometer (MS). In each TPD experiment, the catalyst sample (50 mg) was first treated at 550 °C for 1 h at a ramping rate of 10 °C min⁻¹ in flowing 10% $O_2$/He (50 mL min⁻¹). After cooling down to 150 °C, 1% $NH_3$/He (30 mL min⁻¹) was introduced to the sample until saturation. The sample was then purged in flowing He at 30 mL min⁻¹ for 1 h to remove weakly adsorbed molecules. Subsequently, the sample was ramped to 800 °C at a rate of 10 °C min⁻¹ in flowing He (30 mL min⁻¹), and desorption at 17 amu was recorded by MS.

Single-pulse $^{27}Al$ MAS NMR experiments were performed at 25 °C on a Varian-DDR 14.1 T (with a $^{27}Al$ Larmor frequency of 156.291 MHz) NMR spectrometer using a commercial 1.6 mm pencil-type probe with a rotor spinning rate of 32 kHz. The typical parameters for acquiring NMR spectra for quantitative analysis were spectrum width = 125 kHz, recycle delay time = 3 s (an array of recycle delay time from 0.5 s to 5 s confirmed that 3 s is sufficient for all species reaching equilibrium state between each scan), acquisition time = 20 ms, number of scans = 10240, and a small tip angle π/20 (corresponding to a pulse width of 0.2 μs). Chemical shifts were referenced to 1.0 M $AlCl_3$ aqueous solution at 0 ppm.

## Operando EPR spectroscopy

In addition to quantification of isolated $Cu^{II}$ ions, EPR was also used in situ during catalyst dehydration[53], and operando during steady-state standard SCR[56]. In such measurements, ~10 mg of the sample was placed between two quartz wool plugs in a 3 mm quartz tube with a 0.5 mm hole in the bottom. The loaded tube was positioned in an outer 5 mm sealed quartz tube with an inlet and outlet, making it a plug flow reaction system. Four gas lines with mass flow controllers were connected to a diversion valve just outside the magnet of the EPR spectrometer. During in situ dehydration, a $N_2$ flow of 20 mL min⁻¹ was used. The sample was heated to 350 °C at 10 °C min⁻¹ using a Bruker ER 4131 continuous flow temperature control system, and maintained at this temperature for ~30 min for complete dehydration. After EPR spectrum acquisition at 350 °C, the sample was cooled to −150 °C by liquid $N_2$ before acquiring another spectrum. During operando SCR, a gas mixture containing 350 ppm $NH_3$, 350 ppm NO (containing ~10 ppm $NO_2$), 10% $O_2$, and balance $N_2$ was used. The total gas flow was kept at 100 mL min⁻¹, corresponding to a GHSV of ~$4 \times 10^5$ h⁻¹. SCR was carried out between 350 to 100 °C at 20 °C intervals. At each target temperature, EPR spectrum was acquired after steady state was achieved. The EPR spectra were collected with the magnetic field swept between 2400 and 3600 G in 83 s with a time constant of 83 ms. The field was modulated at 100 kHz and with an amplitude of 5 G, and the microwave frequency was typically 9.3 GHz with a power of 0.2 mW. To quantify EPR active $Cu^{II}$ under operando conditions, the same spectrum double-integration method described above was adopted. An additional correction was made here: as EPR measurements conducted at different temperatures follow a Boltzmann distribution (i.e., peak areas are inversely proportional to the measurement temperature in kelvin)[57], this temperature correction was also incorporated during our quantification. For measurements carried out above SCR onset temperatures, however, quantification uncertainties are difficult to estimate since it is not possible to isolate signal loss from $Cu^{I}$-ion formation, from high $Cu^{II}$-ion mobility, and from measurement error.

## Computational details

All periodic DFT calculations were employed the mixed Gaussian plane wave scheme as implemented into the QUICKSTEP module in the CP2K code[58]. Core electrons were represented with norm-conserving Goedecker-Teter-Hutter pseudopotentials with a plane wave cutoff energy of 360 Ry[59–61]. The generalized gradient approximation (GGA) exchange-correlation functional of Perdew, Burke, and Enzerhof (PBE)[62] was used. Each reaction state configuration was optimized with the Broyden-Fletcher-Goldfarb-Shanno (BGFS) algorithm with SCF convergence criteria of $1.0 \times 10^{-8}$ a.u. The DFT-D3 scheme[63] with an empirical damped potential term was added into the energies obtained from exchange-correlation functional in all calculations, in order to compensate the long-range van der Waals dispersion interactions between the adsorbate and the zeolite. Transition states of elementary steps in the reaction routes were located using the climbing image nudged elastic band (CI-NEB) method[64,65] with seven intermediate images along the reaction pathway between the initial and the final states. Each identified transition state was further confirmed by the vibrational frequency analysis. The vibrational frequencies of the molecules were calculated in the harmonic oscillator approximation with a displacement of 0.01 Å. Only the atoms in the mobile reactants and active site were considered while the other framework atoms of the zeolite were fixed.

Gibbs free reaction energy along reaction pathways were calculated using classical statistical mechanics method[66] including zero-point energy (ZPE) and entropy:

$$G = E_{elec} + E_{ZPE} - TS \qquad (3)$$

where $E_{elec}$ is the electronic energy term, $E_{ZPE}$ is the zero-point energy contribution, S is the entropy, and $T$ is the temperature. $E_{elec}$ was directly derived from DFT calculations. The ZPE contribution is given by:

$$E_{ZPE} = \sum_i \frac{h\nu_i}{2} \qquad (4)$$

where $h$ and $\nu_i$ are the Plank's constant and calculated vibrational frequencies. The entropy term can be calculated as following:

$$-TS = k_B T \sum_i \left( \frac{h\nu_i}{k_B T(e^{h\nu_i/k_B T} - 1)} - \ln\left(1 - e^{-h\nu_i/k_B T}\right) \right) \qquad (5)$$

where $k_B$ is the Boltzmann constant. In the present work, a normal mode of $50 cm^{-1}$ was adopted to replace all the imaginary and low-lying vibrational frequencies in the vibrational entropy calculation. This estimation method has been proposed and widely used in the calculation of vibrational entropy[66–69].

The periodic CHA zeolite model was prepared using two hexagonal unit cells with the size parameters of $13.6750 \times 23.6858 \times 14.7670 Å^3$. To mimic the experimental Si/Al ratio of -15 for CHA zeolite used in this work, four Si atoms on the CHA framework were replaced by 4 Al atoms, and compensating the charges with additional four H atoms on the O atoms, resulting in the model CHA zeolite of $H_4Al_4Si_{68}O_{144}$ with the Si/Al of 17[42,44,69,70]. In particular, two Al atoms at the 8MR between the two *cha* cages were assigned, forming two BASs at the 8MR.

The LTA zeolite structure was modeled using two cubic unit cells (48 T-atoms) with the size parameters of $11.7240 \times 11.7640 \times 23.4460 Å^3$. Three Si atoms within the LTA model structure were replaced by three Al atoms, obtaining a model structure with Si/Al -15. Similarly, two BAS sites at the 8MR were set up for studying the diffusion mechanism of $Cu^{II}(OH)(NH_3)_3$ intermediate.

To simulate hydrolysis and intercage diffusion of CuII-amine complexes, one $Cu^{II}$-ion was introduced to each CHA or LTA unit cell described above, leading to Cu/Al ratios of 0.25 and 0.33, respectively. These values are very close to the Cu/Al ratios of the Cu/CHA-a and Cu/LTA catalysts used in the experimental studies.

## Data availability
All data are available within the paper and its Supplementary Information files or from the corresponding authors upon request.

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

## Acknowledgements

The authors from Pacific Northwest National Laboratory (PNNL) gratefully acknowledge the US Department of Energy (DOE), Energy Efficiency and Renewable Energy, Vehicle Technologies Office for the support of this work. Part of the research described in this paper was performed in the Environmental Molecular Sciences Laboratory (EMSL), a national scientific user facility sponsored by the DOE's Office of Biological and Environmental Research and located at PNNL. PNNL is operated for the US DOE by Battelle. The work at POSTECH was supported by the National Creative Research Initiative Program (2021R1A3A3088711, to S.B.H.) through the National Research Foundation of Korea.

## Author contributions

Y.Q.W., S.H.A., M.A.D., and F.G. prepared the zeolite supports and synthesized the catalysts. Y.Q.W., Y.L.W., and F.G. carried out catalytic reaction tests. Y.Q.W., F.G., and E.D.W. performed EPR measurements and data analysis. Y.C. and N.M.W. performed NMR measurements and data analysis. W.R.Z. and D.H.M. conducted DFT calculations and data analysis. K.G.R. and Y.W. cooperated with discussions and provided valuable suggestions. F.G., K.G.R., Y.W., D.H.M., and S.B.H. raised funds for the research. F.G. and D.H.M. drafted the manuscript and S.B.H. contributed to extensive modifications. All the coauthors contributed to the discussion and helped to revise the manuscript.

## Competing interests

The authors declare no competing interests.
