## [Peer Review File · Nature Communications]

Interplay between copper redox and transfer and support acidity and topology in low temperature NH₃-SCRREVIEWER COMMENTS

Reviewer #1 (Remarks to the Author):

This communication presents and discusses results from operando EPR, kinetic tests and DFT to assess the impact of Bronsted acidity, SAR and topology of the zeolite on the redox activity of Cu-zeolite catalysts for NH₃-SCR: in this respect, the paper fills a gap and is therefore welcome. Also, the adopted methods are innovative and effective, and of the highest quality. A few aspects deserve however some consideration before the manuscript can be published in Nat. Comm.

1) The results in this paper point to the RHC as the rate controlling step of the Standard SCR redox mechanism. Accordingly, the authors ascribe the observed kinetic relevance of BAS to its impact on the rates of i) the hydrolysis of the redox resistant CuII(NH₃)₄ complex to give the redox active CuII(OH)(NH₃)₃, and ii) the intercage transport of CuII(OH)(NH₃)₃ to form the dinuclear CuII complexes active in RHC. In principle, this makes sense: however, I have the following objection. Gramigni et al. <https://doi.org/10.1021/acscatal.0c05362> and recently Daya et al. <https://doi.org/10.1021/acscatal.2c01076> have independently shown that the RHC rate over Cu/CHA is 2nd order in [CuII]. On the other hand, both the hydrolysis of CuII(NH₃)₄ and the intercage transport of CuII(OH)(NH₃)₃ are clearly 1st order processes. So, how can they be kinetically relevant? I wonder if BAS could affect instead the thermodynamics of those two steps, which precede the consecutive formation of the dinuclear CuII species involved in the RHC mechanism and may be quasi-equilibrated, the slow step being the subsequent activation of NO on the dinuclear CuII complex. The authors may want to discuss this point and add the two relevant references above.

2) Along similar lines, the lower Std. SCR activity of Cu/CHA-b in comparison to Cu/CHA-a and Cu/AEI (Figure 1 and Supplementary Figure 2) can be more likely ascribed to its lower Cu density hindering the 2nd order RHC kinetics.

3) While the steady-state Std. SCR data of Supplementary Figure 2 were collected under wet conditions (2.5 % H₂O), the EPR operando experiments were run under dry conditions. H₂O is known to inhibit RHC and promote OHC (Ottinger et al. / SAE Int. J. Advances & Curr. Prac. in Mobility, Volume 3, Issue 6, 2021; also, <https://doi.org/10.1021/acscatal.0c05362>): the impact of H₂O should be added to the discussion of the correlation between these two sets of measurements.

4) All the discussions in the paper refer essentially to the RHC of the SCR redox mechanism, focusing on the hydrolysis and mobility of CuII ions. The RHC is not necessarily always the rate determining step, though: see e.g. the SAE paper mentioned above, Figure 1, which shows that the H₂O effect on the Std. SCR activity changes from negative to positive as the NO feed concentration is increased, due to OHC becoming the slower half cycle. So, I think that the authors should at least add a warning to their discussion, mentioning that their results and conclusions refer to specific SCR conditions where the RHC is rate controlling. What would be the impact of e.g. BAS when the OHC becomes rate limiting?

Reviewer #2 (Remarks to the Author):

The manuscript “Low-temperature NH₃-SCR over copper-exchanged small-pore zeolites: An interplay between copper redox and transfer, and support acidity and topology” by Gao et. al. describes the results of applying operando EPR techniques and DFT calculations on copper zeolite materials. These materials have been under intense scrutiny by many groups using a multitude of experimental and computational methods, including EPR methods measurements. The Authors here use an operando EPR set-up monitoring the Cu(II) structure and dynamics under reaction conditions. The amount of EPR active sites is found to correlate inversely with the catalytic efficiency. This behavior is rationalized by the Authors considering literature proposed mechanisms related to the conversion of tetraminocopper(II) species into triaminohydroxocopper(II) moieties. Differences in the behavior of CHA and LTA zeolites is linked to the different structure and Brønsted sites populations in the two materials. The work points to the importance of Brønsted acid sites and on their dependence on the different framework structures investigated. One major concern is that the Si/Al and Cu/Al ratios are all different, thus making such a conclusion unreliable. The spectroscopic work relies on a series of very nice CW-EPR spectra taken in the temperature interval 50-150°C and low T spectra. The interpretation of the spectra is however somewhat disappointing. Since the Authors measured the rigid limit spectra at low T, computer simulations should be given and the full tensor elements provided. With these in hand the averaged high T spectrum should be simulated using the same spin Hamiltonian parameters this obtaining the mobility of the solvated Cu species and estimates of the hydrodynamic radius. This approach has been reported recently in *Nat Commun* 12, 4638 (2021), which should be cited. This approach will also allow to obtain a more reliable quantitative estimates of the differ species monitored by EPR. To this end it is surprising that no reference is made to the formation of mixed-ligand mobile [CuII(NH₃)₃(NO₃)]⁺ complexes in the operando conditions recently reported (*ChemCatChem*, 11(16), pp.3828-3838.). These should be EPR active and given the steric hindrance should be characterized by slower rotational correlation times, which may be reflected in the different line shapes observed in the spectra reported in Figure 3. Significant hints should therefore come from the accurate simulation analysis of the reported EPR spectra.

Another major concern is related to the quantitative aspects related to spectra taken at high temperature. At page 5 a link is proposed between the amount of Cu(II) active species and catalytic activity related to Cu(II) to Cu(I)-EPR silent- species. Here the Authors should be very careful and explain accurately this correlation. Infact increased mobility at 150° leads to faster relaxations and part of Cu(II) may be EPR inactive. Moreover, at page 4 they say “that dehydrated ZCuII(OH) is EPR silent due to a pseudo Jahn-Teller effect”. This statement has been recently proven to be wrong and the EPR signature of these species has been reported (*J. Am. Chem. Soc.* 2022, 144, 29, 13079–13083). The only concern is that the investigation has not revealed any real breakthrough in the knowledge about the materials. Therefore, the manuscript should be judged by the significance of the results, namely, did the work give any new insight on any catalytic aspect of Cu(II) in Chabazite ? This is not obvious. The Authors should

elaborate on this carefully clarifying the above reported concerns.

Minor points

EPR data for molecular tetraminocopper(II) species should be given for the sake of comparison.

Page 6 line 267 “ It is also remarkable that NH₃ desorption from BASs greatly decreases after hydrothermal aging..” why is this remarkable? It is what expected as a consequence of Al leaching and reduction of H Broensted sites.

Pag 8 line 327 “high-field region signals also display some clearly resolved superhyperfine features. “ The mentioned superhyperfine features are not clearly resolved at all. They should be clearly indicated and analyzed.

DFT methods. The number of Cu per cage should be given and compared with the experimental data.

Reviewer #3 (Remarks to the Author):

The manuscript can be accepted for publication in the present form.

Response to reviewer comments:

Reviewer #1 (Remarks to the Author):

This communication presents and discusses results from operando EPR, kinetic tests and DFT to assess the impact of Bronsted acidity, SAR and topology of the zeolite on the redox activity of Cu-zeolite catalysts for NH₃-SCR: in this respect, the paper fills a gap and is therefore welcome. Also, the adopted methods are innovative and effective, and of the highest quality. A few aspects deserve however some consideration before the manuscript can be published in Nat. Comm.

The reviewer's positive remarks are much appreciated.

1) The results in this paper point to the RHC as the rate controlling step of the Standard SCR redox mechanism. Accordingly, the authors ascribe the observed kinetic relevance of BAS to its impact on the rates of i) the hydrolysis of the redox resistant Cu^{II}(NH₃)₄ complex to give the redox active Cu^{II}(OH)(NH₃)₃, and ii) the intercage transport of Cu^{II}(OH)(NH₃)₃ to form the dinuclear Cu^{II} complexes active in RHC. In principle, this makes sense: however, I have the following objection.

Gramigni et al. <https://doi.org/10.1021/acscatal.0c05362> and recently Daya et al.

<https://doi.org/10.1021/acscatal.2c01076> have independently shown that the RHC rate over Cu/CHA is 2nd order in [Cu^{II}]. On the other hand, both the hydrolysis of Cu^{II}(NH₃)₄ and the intercage transport of Cu^{II}(OH)(NH₃)₃ are clearly 1st order processes. So, how can they be kinetically relevant? I wonder if BAS could affect instead the thermodynamics of those two steps, which precede the consecutive formation of the dinuclear Cu^{II} species involved in the RHC mechanism and may be quasi-equilibrated, the slow step being the subsequent activation of NO on the dinuclear Cu^{II} complex. The authors may want to discuss this point and add the two relevant references above.

The reviewer's insights into low-temperature SCR mechanisms are much appreciated. We are also thankful to the reviewer for providing the relevant references. The reviewer started this comment by stating that "the RHC as the rate controlling step of the Standard SCR redox mechanism". We can certainly understand why the reviewer has such an impression: the EPR technique is only sensitive to Cu^I species and Cu^I species are obviously associated with RHC. Since both isolated Cu^I-ions and any Cu-dimer species (irrespective of Cu oxidation state) are EPR silent, operando EPR is incapable of addressing OHC. Furthermore, our kinetics and operando EPR reported here were all measured under steady-state conditions, i.e., we did not purposely isolate RHC and OHC in our studies. However, we don't believe that RHC is the sole rate-limiting step under most practically relevant SCR conditions. RHC and OHC can both have nonnegligible kinetic influences. As a quasi-homogeneous process, another key parameter that can profoundly influence kinetics is Cu transfer. It is readily understood that faster Cu migration benefits both RHC and OHC. In our recent publication, these aspects were addressed in detail¹.

Regarding the 2nd order in [Cu^{II}] for RHC rate over Cu/CHA, we would like to comment that both the Tronconi group^{2,3} and the Cummins + ORNL group⁴ derived this concept from transient NH₃+NO titration kinetic analysis. This concept clearly has its value; however, it is not fully consistent with kinetic behavior derived from steady-state measurements where over a wide range of Cu density, SCR rates increase *linearly* with Cu loading (i.e., invariant turnover rates)^{5,6}. We can understand this discrepancy, at least qualitatively, from the notion that transient NH₃+NO titration works on all reducible Cu^{II}-ions irrespective of their relative reactivity, but steady-state kinetics are likely contributed most by Cu^{II}-ions

that are the most reactive. Since Cu-ions in Cu/CHA follow non-mean field kinetic behavior⁷, global kinetic parameters derived by titrating all Cu-ions certainly have their limitations.

Figure 4 from Ref. 2 and Figure 5 from Ref. 4 are adapted below (these plots were key for the authors to claim the 2nd order dependence). From the Tronconi group data, 2nd order simulation fits the experimental data better, but 1st order simulation is not completely off the mark; it still provides a decent kinetic description. For the Cummins + ORNL group data, 2nd order simulation did not even mimic experimental data that well. Even though the authors didn't provide a 1st order simulation, we are sure that if they did, it will also reasonably mimic experimental data.

Figure 4 from Ref. 2

Figure 5 from Ref. 4

From the discussions above, we don't believe that it is fair to state that the 2nd order in [Cu^{II}] for RHC derived under transient titration conditions can be used to preclude kinetic relevance of the hydrolysis of Cu^{II}(NH₃)₄ and the interstage transport of Cu^{II}(OH)(NH₃)₃ under steady-state reaction conditions. Rather, our operando EPR data clearly demonstrate that when the support Brønsted acid strength is lowered (i.e., by using LTA rather than the more acidic CHA or AEI supports), or the support Brønsted acid density is lowered (i.e., by hydrothermal treatment), Cu^{II}(NH₃)₄ becomes much more detectable (i.e., much less reactive). This is compelling evidence that the hydrolysis of Cu^{II}(NH₃)₄ and the interstage transport of Cu^{II}(OH)(NH₃)₃ can certainly slow down SCR rates when help from support Brønsted acidity is insufficient.

To address this important comment, on page 6 of the revised manuscript, the second paragraph has been modified as follows:

It is important to note from the calculations above that the energy barriers for the hydrolysis and interstage diffusion of the associated Cu^{II} -amine complexes are around the same magnitude as standard SCR activation energies (typically 60-80 kJ/mol). As such, the hydrolysis and interstage diffusion processes can certainly play kinetically relevant roles, i.e., the redox chemistry does not always play sole rate-limiting roles during low-temperature standard SCR. The low-temperature SCR kinetics, *operando* EPR and DFT studies presented above clearly show the kinetic relevance of zeolite BAS strength, i.e., at low BAS strength, $\text{Cu}^{\text{II}}(\text{NH}_3)_4$ hydrolysis to $\text{Cu}^{\text{II}}(\text{OH})(\text{NH}_3)_3$ is no longer so facile as not to display any kinetic consequences.

2) Along similar lines, the lower Std. SCR activity of Cu/CHA-b in comparison to Cu/CHA-a and Cu/AEI (Figure 1 and Supplementary Figure 2) can be more likely ascribed to its lower Cu density hindering the 2nd order RHC kinetics.

We agree with the reviewer on this comment.

On page 3 of the revised manuscript, a modification has been made:

Following this rationale, the lower atomic efficiency for Cu/CHA-b as compared to Cu/CHA-a and Cu/AEI is readily attributed to its lower overall Cu ion content (rather than its $\text{Z}_2\text{Cu}^{\text{II}}$ vs. $\text{ZCu}^{\text{II}}\text{OH}$ speciation). With such an active site concentration decrease, the degree of rate control from Cu ion transfer increases, but that from the intrinsic redox chemistry decreases.

3) While the steady-state Std. SCR data of Supplementary Figure 2 were collected under wet conditions (2.5 % H_2O), the EPR *operando* experiments were run under dry conditions. H_2O is known to inhibit RHC and promote OHC (Ottinger et al. / SAE Int. J. Advances & Curr. Prac. in Mobility, Volume 3, Issue 6, 2021; also, <https://doi.org/10.1021/acscatal.0c05362>): the impact of H_2O should be added to the discussion of the correlation between these two sets of measurements.

This valuable comment is well received. For technical reasons, our *operando* EPR system is shared with other research projects which precluded us from the addition of a water saturator to the system. As such, the *operando* EPR measurements were only carried out dry.

From the references provided by the reviewer, it is clear that H_2O influences RHC and OHC much more profoundly under transient titration than steady-state conditions. In the presence of H_2O , the RHC E_{act} was calculated to be 25-30 kJ/mol². These energies are far away from typical low-T standard SCR activation energies (60-80 kJ/mol), but are very close to hydrogen bonding energies between H_2O molecules (~ 25 kJ/mol). We have good reason to suggest, therefore, that the wet E_{act} reported in Ref² may not reflect intrinsic RHC activation. In order to explain this apparently unrealistically low energy, Tronconi and coworkers pointed out in Ref² that “However, it is well-known that H_2O does not significantly affect the steady state deNO_x activity under Standard SCR conditions: this is apparently at variance with the above results showing a strong negative impact of H_2O on the RHC rate. Preliminary results from our parallel ongoing study of the OHC kinetics suggest in fact a positive H_2O effect on the Cu^{I} oxidation rate, which can offset the negative H_2O effect on the RHC at steady-state Standard SCR conditions”, which we tentatively agree. Since the results presented in the current study were only collected under steady state, we feel

that other than clearly stating key literature findings under transition measurements, no further discussions of the H₂O effect is necessary.

In the revised manuscript, the following description has been added on page 4:

...We use the same approach here to compare the current catalysts. We note that ~2.5% cofed H₂O was applied in our kinetic measurements shown above, but not during our *operando* EPR measurements. Even though cofed H₂O has been reported to markedly influence transient SCR kinetics, its influence under steady state (applied here) is rather minimal^{2,8}. As such, good correlations between our kinetics and spectroscopic studies are anticipated. The *operando* EPR spectra...

4) All the discussions in the paper refer essentially to the RHC of the SCR redox mechanism, focusing on the hydrolysis and mobility of Cu^{II} ions. The RHC is not necessarily always the rate determining step, though: see e.g. the SAE paper mentioned above, Figure 1, which shows that the H₂O effect on the Std. SCR activity changes from negative to positive as the NO feed concentration is increased, due to OHC becoming the slower half cycle. So, I think that the authors should at least add a warning to their discussion, mentioning that their results and conclusions refer to specific SCR conditions where the RHC is rate controlling. What would be the impact of e.g. BAS when the OHC becomes rate limiting?

This review comment is well received. As discussed earlier, we fully agree with the reviewer that “RHC is not necessarily always the rate determining step”. However, under most practically relevant catalyst composition and reaction conditions, an OHC dominance is rare. That said, kinetic dominance from OHC can be purposely controlled, e.g., by dramatically lowering Cu concentration¹, or by lowering O₂ pressure during reaction⁹. Under such extreme conditions, it is anticipated that the BAS influence will become less significant since BAS mainly affects RHC.

On page 8 of the revised manuscript, the following paragraph has been added:

Finally, regarding the interplay between RHC and OHC in low-temperature SCR, both half-cycles contribute to rate-controlling in a convoluted manner under practically relevant catalyst composition and reaction conditions during steady-state operations.⁹ Because of the EPR silent nature of Cu^I-ions and dimeric Cu intermediates that are involved in OHC, *operando* EPR cannot be applied to address this half-cycle. Nevertheless, kinetic dominance from OHC can be purposely designed, e.g., by dramatically lowering Cu concentration¹, or by lowering O₂ pressure during SCR reaction⁹. Under such extreme conditions, it is anticipated that the BAS influence will be less significant since BAS mainly affects RHC.

Reviewer #2 (Remarks to the Author):

The manuscript “Low-temperature NH₃-SCR over copper-exchanged small-pore zeolites: An interplay between copper redox and transfer, and support acidity and topology” by Gao et. al. describes the results of applying *operando* EPR techniques and DFT calculations on copper zeolite materials. These materials have been under intense scrutiny by many groups using a multitude of experimental and computational methods, including EPR methods measurements. The Authors here use an *operando* EPR set-up monitoring the Cu(II) structure and dynamics under reaction conditions. The amount of EPR active sites is found to correlate inversely with the catalytic efficiency. This behavior is rationalized by

the Authors considering literature proposed mechanisms related to the conversion of tetraminocopper(II) species into triaminohydroxocopper(II) moieties. Differences in the behavior of CHA and LTA zeolites is linked to the different structure and Bronsted sites populations in the two materials. The work points to the importance of Bronsted acid sites and on their dependence on the different framework structures investigated.

The positive remarks from the reviewer are much appreciated.

One major concern is that the Si/Al and Cu/Al ratios are all different, thus making such a conclusion unreliable.

We understand the reviewer's concerns, but we don't believe this is a fair comment. From a zeolite synthesis point of view, CHA can be readily synthesized with varying Si/Al ratios, but it is not yet the case for AEI and LTA synthesis under the current state of the art. As such, we synthesized CHA-a and CHA-b to match Si/Al ratios of LTA and AEI, respectively. Regarding Cu loading, we used solution ion exchange to obtain catalysts with intermediate Cu/Al ratios to avoid the presence of multinuclear Cu moieties to ease our kinetics and EPR studies. Overall, our comparative studies were not based on randomly chosen catalyst samples.

In any case, to address the reviewer's comment, we synthesized two new sets of Cu/CHA-a and Cu/LTA samples (note that CHA-a and LTA have very similar Si/Al ratios) using a so-called solid-state ion exchange method¹⁰, where Cu loadings (1-4 wt%) are under much better control than solution ion exchange. Using these new samples, we carried out SCR kinetics and operando EPR measurements. Such studies will be reported elsewhere; herein some representative results are shown for the purpose of review only.

From Table 1 shown below, the two series of Cu/LTA and Cu/CHA catalysts have very similar Si/Al and isolated Cu/Al ratios. As such, these samples are more comparison friendly as compared to the catalysts used in the present study.

Table 1: total Cu and isolated Cu (from EPR) contents of the Cu/LTA and Cu/CHA catalysts.

Catalyst Name	Cu loading (wt%)	Isolated Cu ^{II} content (wt%, EPR)	Catalyst Name	Cu loading (wt%)	Isolated Cu ^{II} content (wt%, EPR)
Cu/LTA-1	1.0%	0.80%	Cu/CHA-1	1.0%	0.87%
Cu/LTA-2	2.0%	1.55%	Cu/CHA-2	2.0%	1.57%
Cu/LTA-3	3.0%	1.69%	Cu/CHA-3	3.0%	1.58%
Cu/LTA-4	4.0%	1.68%	Cu/CHA-4	4.0%	1.72%

From the Arrhenius plots shown below, the newly prepared Cu/CHA samples are still more active than the corresponding Cu/LTA samples, and from the operando EPR spectra, the Cu/LTA samples still display stronger isotropic Cu^{II}(NH₃)₄ signals, e.g., at 150 and 225 °C, than their Cu/CHA counterparts. Again,

these new data will be published elsewhere, and we do not intend to provide further analysis here. But these new data serve to fully support reproducibility of our kinetic and operando EPR data used in the present study, and the main conclusions drawn based on such data.

Figure 1 (left) Arrhenius plots of SCR TORs vs. $1/T$; (middle) selected operando spectra over Cu/LTA at selected temperatures; (right) selected operando spectra over Cu/CHA at selected temperatures.

The spectroscopic work relies on a series of very nice CW-EPR spectra taken in the temperature interval 50-150°C and low T spectra. The interpretation of the spectra is however somewhat disappointing. Since the Authors measured the rigid limit spectra at low T, computer simulations should be given and the full tensor elements provided. With these in hand the averaged high T spectrum should be simulated using the same spin Hamiltonian parameters this obtaining the mobility of the solvated Cu species and estimates of the hydrodynamic radius. This approach has been reported recently in Nat Commun 12, 4638 (2021), which should be cited. This approach will also allow to obtain a more reliable quantitative estimates of the differ species monitored by EPR.

We appreciate the reviewer's suggestions on conducting spectrum simulations to describe Cu species in a more quantitative way. We are also thankful to the reviewer for providing the Nat. Comm. reference. The Nat. Comm. paper described hydration-dehydration of Cu/CHA. With the aid from HYSORE, ^{17}O ENDOR and DFT, the authors were able to quantitatively describe $[\text{Cu}^{\text{II}}(\text{H}_2\text{O})_6]$ and $[\text{Cu}^{\text{II}}(\text{H}_2\text{O})_4(\text{O}-8\text{MR})_2]$ species, which display isotropic and anisotropic EPR characteristics, respectively, under ambient conditions. For the samples studied here, however, these assignments appear incomplete. Our H_2 -TPR results shown in Supplementary Figure 3 demonstrate the presence of ZCuOH in all of our samples. This is not surprising. EPR parameters for isotropic $[\text{Cu}^{\text{II}}(\text{H}_2\text{O})_6]$ and $[\text{Cu}^{\text{II}}(\text{OH})(\text{H}_2\text{O})_5]$ are likely very similar and difficult to distinguish.

In the revised manuscript, the following description has been added on page 4 that reads:

By ^{17}O isotopic labelling of the zeolite framework, in conjunction with advanced EPR methodologies and DFT modelling, Bruzzese et al.¹¹ recently attributed the isotropic ions to $[\text{Cu}^{\text{II}}(\text{H}_2\text{O})_6]$ and anisotropic ions to $[\text{Cu}^{\text{II}}(\text{H}_2\text{O})_4(\text{O}_\text{L})_2]$ where O_L denotes lattice oxygen of the zeolite support. We note that such attributions are most certainly incomplete for the samples used here. For example, the presence of mobile $[\text{Cu}^{\text{II}}(\text{OH})(\text{H}_2\text{O})_x]$ in our samples is clearly evidenced by H_2 -TPR results in Supplementary Fig. 3.

Operando EPR during SCR, unfortunately, is much more complex to deal with via spectrum simulations. The "the rigid limit spectra at low T" shown in Figure 2 of the manuscript unfortunately does not apply to Cu^{II} -amine species in Figures 3 and 7 for two reasons: (1) such species follow dynamic temperature-

dependent evolution (e.g., the ligand number n decreases with increasing temperature), (2) as Cu-N coordination number decreases, Cu-O_L number increases. As such, the spin Hamiltonian parameters do not maintain at different temperatures. For example, at a reaction temperature of 150 °C, we did not observe [Cu^{II}(NH₃)₄] and [Cu^{II}(OH)(NH₃)₃] at all on Cu/CHA and Cu/AEI samples; the only detected species are [Cu^{II}(NH₃) _{n} (O_L)_{4- n}] (Figure 3b of the manuscript). As such, we only provided “snapshots” at 4 different temperatures in Figure 3, and only described these spectra in a qualitative manner.

We indeed carried out some spectra simulations as the reviewer suggested. As an example, we did obtain parameters from a high temperature spectrum (Cu/CHA-a dehydrated at 350 °C and presumable therefore had minimal motion), and simulated motional spectra using these parameters at a range of rotational diffusion rates. This figure is shown below:

Figure 2. Simulated Cu EPR spectra showing the effect of motion. Spin Hamiltonian parameters were first obtained by fitting the 350 °C spectra of Cu/CHA-a. Then using these parameters, a series of spectra were simulated with the addition of rotational motion, ranging from Log 7.5 to 9.1 s⁻¹.

From the example above, by allowing motion of dehydrated (and thus immobilized) Cu, lineshape of hydrated and mobile Cu spectra can indeed be generated. The success of this simulation lies in the fact that the nature of ligand (in this case Cu-O) does not change. In attempting to simulate our operando spectra that contain both N and O_L ligands, we did not obtain any meaningful results.

In any case, this review comment has prompted us to plan future experiments where we attempt to capture various Cu-N coordinated and Cu-(N, O_L) coordinated species by flash freezing samples under operando conditions to obtain their rigid limit parameters.

To this end it is surprising that no reference is made to the formation of mixed-ligand mobile [Cu^{II}(NH₃)₃(NO₃)⁺ complexes in the operando conditions recently reported (ChemCatChem, 11(16), pp.3828-3838.). These should be EPR active and given the steric hindrance should be characterized by slower rotational correlation times, which may be reflected in the different line shapes observed in the spectra reported in Figure 3. Significant hints should therefore come from the accurate simulation analysis of the reported EPR spectra.

We are aware of this study by Negri et al.¹², and cited it in our recent work¹. Here we would like to make two arguments: first, the $[\text{Cu}^{\text{II}}(\text{NH}_3)_3(\text{NO}_3)]^+$ complex forms under a specific condition, i.e., the generation of $\text{Cu}-\text{NO}_3^-$ first, followed by NH_3 exposure at temperatures below that NH_3 interacts with NO_3^- . It is unlikely that the same species forms under typical steady-state low-temperature SCR conditions. This follows since a nitrate (or nitrite) ligand less likely can compete favorably with NH_3 ligands at steady state. Indeed, as Marberger et al.¹³ pointed out in their XAS studies that “ $\text{Cu}^{\text{II}}-(\text{NO}_x)_y$ -type species were observed, but only when the catalyst was no longer fully covered by NH_3 and after the disappearance of gas-phase NH_3 .” Second, even if small quantities of $[\text{Cu}^{\text{II}}(\text{NH}_3)_3(\text{NO}_3)]^+$ complex does form under our operando conditions at 50 °C, as a mobile complex, it will not generate any split hyperfine signals. It is only anticipated that its g_{ave} signal will be completely overwhelmed by the much stronger $[\text{Cu}^{\text{II}}(\text{NH}_3)_4]$ and $[\text{Cu}^{\text{II}}(\text{OH})(\text{NH}_3)_3]$ g_{ave} signals. At higher temperatures, its presence will be even more questionable considering its reactivity. Note again from Figure 3b that at 150 °C, no mobile Cu-amine complexes were detected on Cu/AEI and Cu/CHA.

To address this comment, on page 5 of the revised manuscript, the following descriptions have been introduced that read:

Note that at the same temperature of 50 °C, Negri et al.¹² demonstrated the formation of a mobile $[\text{Cu}^{\text{II}}(\text{NH}_3)_3(\text{NO}_3)]^+$ complex by generating $\text{Cu}^{\text{II}}-\text{NO}_3^-$ first, followed by NH_3 ligating. Under steady-state conditions applied here, the presence of this mixed-ligand species, however, is not very likely. Indeed, as Marberger et al.¹³ pointed out in their XAS studies that $\text{Cu}^{\text{II}}-(\text{NO}_x)_y$ -type species were observed, but only when the catalyst was no longer fully covered by NH_3 and after the disappearance of gas-phase NH_3 . Even if small quantities of $[\text{Cu}^{\text{II}}(\text{NH}_3)_3(\text{NO}_3)]^+$ complex does form under our operando conditions at 50 °C, as a mobile complex, it will not generate any split hyperfine signals. It is only anticipated that its g_{ave} signal will be completely overwhelmed by the much stronger $[\text{Cu}^{\text{II}}(\text{NH}_3)_4]$ and $[\text{Cu}^{\text{II}}(\text{OH})(\text{NH}_3)_3]$ g_{ave} signals. As such, no further discussion about this species is given below.

Another major concern is related to the quantitative aspects related to spectra taken at high temperature. At page 5 a link is proposed between the amount of Cu(II) active species and catalytic activity related to Cu(II) to Cu(I)-EPR silent- species. Here the Authors should be very careful and explain accurately this correlation. In fact increased mobility at 150°C leads to faster relaxations and part of Cu(II) may be EPR inactive.

We thank the reviewer for this excellent comment. It is certainly possible that faster relaxations render some Cu(II) species EPR invisible. From Figure 3a, it is clear that Cu^{II} -amine complexes in Cu/LTA are more mobile at 50 °C (before SCR lights off). It is not very likely that by 150°C, such complexes in Cu/LTA become less mobile than their counterparts in Cu/CHA and Cu/AEI. Since Cu^{II} -amines in Cu/LTA have the highest EPR visibility under SCR conditions, the correlation we drew on page 5 of the manuscript should still hold even with the likelihood that some Cu(II) are EPR inactive due to fast relaxations.

To address this comment, on page 5-6 of the revised manuscript, the following description has been added that reads:

Note that in using quantitative EPR measurements to correlate Cu atomic efficiency, one possible complexity is that some highly mobile Cu^{II} moieties may become EPR silent due to fast

relaxations. However, since our ex-situ and operando EPR spectra shown above suggest higher mobility of Cu^{II} -ions in LTA, the trend shown immediately above will hold even with this uncertainty.

Moreover, at page 4 they say “that dehydrated $\text{ZCu}^{\text{II}}\text{OH}$ is EPR silent due to a pseudo Jahn-Teller effect”. This statement has been recently proven to be wrong and the EPR signature of these species has been reported (J. Am. Chem. Soc. 2022, 144, 29, 13079–13083).

We thank the reviewer for providing this reference. We note, however, that even though some $\text{ZCu}^{\text{II}}\text{OH}$ remains EPR active as this new reference demonstrates, the amount is way below that is originally present in a hydrated sample (i.e., before the dehydration treatment). For example, in a Cu/CHA sample with a composition of Si/Al = 12 and Cu/Al = 0.67, the amount of EPR active, dehydrated $\text{ZCu}^{\text{II}}\text{OH}$ is only 16%¹⁴. According to the Cu site compositional phase diagram versus Si:Al and Cu:Al ratios by Paolucci et al.¹⁵, the amount of $\text{ZCu}^{\text{II}}\text{OH}$ prior to dehydration should be easily higher than 50%. As such, this new reference does not provide a good means of reliably quantifying $\text{ZCu}^{\text{II}}\text{OH}$ in samples prior to dehydration. This great mismatch can be due to $\text{ZCu}^{\text{II}}\text{OH}$ dimerization or autoreduction (both chemistries render EPR invisibility), however, we are afraid that it is not fair to completely rule out the possibility that some dehydrated $\text{ZCu}^{\text{II}}\text{OH}$ sites become EPR silent due to pseudo Jahn-Teller effect. For our dehydrated samples shown in Figure 2c, EPR active $\text{ZCu}^{\text{II}}\text{OH}$ is barely detected, yet we still observed marked EPR signal loss after sample dehydration (Figure 2d). We have no experimental evidence to suggest that this loss is due entirely to $\text{ZCu}^{\text{II}}\text{OH}$ dimerization or autoreduction. Instead of saying that the pseudo Jahn-Teller effect is completely wrong, in the revised manuscript, we decided to include this new reference and a few other references on $\text{ZCu}^{\text{II}}\text{OH}$ dimerization or autoreduction chemistries that lead to EPR signal loss.

On page 4 of the revised manuscript, the following changes have been made that read:

Fig. 2c shows the EPR spectra at -150 °C of the dehydrated samples by heating to 350 °C in dry N_2 . Unlike $\text{Z}_2\text{Cu}^{\text{II}}$ and $\text{ZCu}^{\text{II}}\text{OH}$ in hydrated state, dehydrated $\text{ZCu}^{\text{II}}\text{OH}$ has been suggested to be EPR silent due to a pseudo Jahn-Teller effect¹⁶. More recently, Bruzzese et al. demonstrated that dehydrated $\text{ZCu}^{\text{II}}\text{OH}$ is still EPR active (with $g_{\parallel} = 2.29$ and $A_{\parallel} = 114$ G). Since such hyperfine signals are barely detected in our dehydrated samples, we are not in a position to fully rule out the pseudo Jahn Teller effect. Furthermore, $\text{ZCu}^{\text{II}}\text{OH}$ has been known to lose EPR visibility during dehydration via dimerization and autoreduction chemistries^{17,18}. As such, we assign all ESR signals in Fig. 2c to $\text{Z}_2\text{Cu}^{\text{II}}$.

The only concern is that the investigation has not revealed any real breakthrough in the knowledge about the materials. Therefore, the manuscript should be judged by the significance of the results, namely, did the work give any new insight on any catalytic aspect of Cu(II) in Chabazite? This is not obvious. The Authors should elaborate on this carefully clarifying the above reported concerns.

We understand the reviewer’s concern, but we are afraid that we cannot agree with the lack of new insight in the present study. In this study, we demonstrate via operando EPR that $[\text{Cu}^{\text{II}}(\text{NH}_3)_4]$, a widely proposed reaction intermediate for low-T SCR, is much more reluctant to react when (1) the support acid strength is lowered, and (2) the support acid density is lowered. We further use DFT simulations to suggest that this reluctance is related to hydrolysis and interstage transport of $[\text{Cu}^{\text{II}}(\text{NH}_3)_4]$. These new findings provide important molecular-level explanations to (1) activity difference of different Cu-zeolite

SCR catalysts, and (2) catalyst activity loss caused by hydrothermal aging. We firmly believe that these are important catalytic aspects of Cu-zeolite SCR catalysts.

Minor points

EPR data for molecular tetraminocopper(II) species should be given for the sake of comparison.

This suggestion is well received. A free $[\text{Cu}^{\text{II}}(\text{NH}_3)_4]$ ion in solution or in a zeolite host only displays split hyperfine at cryogenic temperatures where its mobility is restrained^{19,20}. However, the spin Hamiltonian parameters thus obtained are not useful for comparison purposes under SCR conditions where $[\text{Cu}^{\text{II}}(\text{NH}_3)_4]$ ions are highly mobile and only display isotropic EPR characteristics.

Page 6 line 267 “It is also remarkable that NH_3 desorption from BASs greatly decreases after hydrothermal aging..” why is this remarkable? It is what expected as a consequence of Al leaching and reduction of H Bronsted sites.

We fully agree with the reviewer that it is commonly found that NH_3 desorption from BAS decreases after hydrothermal aging. In the revised manuscript (page 7), “it is also remarkable that...” has been changed to “note also that...”

Pag 8 line 327 “high-field region signals also display some clearly resolved superhyperfine features. “ The mentioned superhyperfine features are not clearly resolved at all. They should be clearly indicated and analyzed.

The superhyperfine features are the “spikes” in the high-field region. When Figure 7b and 7c are compared, it is clear that the spikes on the 225 °C spectra are much stronger. We don’t understand why the reviewer thinks that these are not clearly resolved at all.

In any case, superhyperfine signals are better presented by taking derivatives of the EPR spectra. Plotted below are the derivatives of the spectra measured at 150 and 225 °C, respectively. Superhyperfine features (marked using a dashed circle) are indeed much stronger and better resolved at 225 °C. These plots are added to the revised SI file as Supplementary Figure 12.

Figure 3. Derivatives of the operando EPR spectra shown Figure 7b (left) and Figure 7c (right) of the main text.

DFT methods. The number of Cu per cage should be given and compared with the experimental data.

In our DFT calculations, LTA and CHA zeolite support unit cell compositions were $H_3Al_3Si_{45}O_{96}$ and $H_4Al_4Si_{68}O_{144}$, respectively. These compositions were determined based on the experimentally derived Si/Al ratios for the LTA and CHA zeolites of 15.8 and 17.0, respectively. For each unit cell, one Cu^{II}-ion was introduced, leading to Cu/Al ratios of 0.33 and 0.25, respectively. Again, these values are almost identical to the actual Cu/Al ratios of the Cu/LTA and Cu/CHA-a catalysts. Such information has been added to the revised Methods section on page 13.

- 1 Wu, Y. Q. *et al.* Rate Controlling in Low-Temperature Standard NH₃-SCR: Implications from Operando EPR Spectroscopy and Reaction Kinetics. *J Am Chem Soc* **144**, 9734-9746, doi:10.1021/jacs.2c01933 (2022).
- 2 Gramigni, F. *et al.* Transient Kinetic Analysis of Low-Temperature NH₃-SCR over Cu-CHA Catalysts Reveals a Quadratic Dependence of Cu Reduction Rates on Cu-II. *Acs Catal* **11**, 4821-4831, doi:10.1021/acscatal.0c05362 (2021).
- 3 Hu, W. S. *et al.* On the Redox Mechanism of Low-Temperature NH₃-SCR over Cu-CHA: A Combined Experimental and Theoretical Study of the Reduction Half Cycle. *Angew Chem Int Edit* **60**, 7197-7204, doi:10.1002/anie.202014926 (2021).
- 4 Daya, R. *et al.* Kinetic Model for the Reduction of Cu(II) Sites by NO + NH₃ and Reoxidation of NH₃-Solvated Cu(I) Sites by O₂ and NO in Cu-SSZ-13. *Acs Catal* **12**, 6418-6433, doi:10.1021/acscatal.2c01076 (2022).
- 5 Gao, F. *et al.* Understanding ammonia selective catalytic reduction kinetics over Cu/SSZ-13 from motion of the Cu ions. *J Catal* **319**, 1-14, doi:DOI 10.1016/j.jcat.2014.08.010 (2014).
- 6 Paolucci, C. *et al.* Dynamic multinuclear sites formed by mobilized copper ions in NO_x selective catalytic reduction. *Science* **357**, 898-903, doi:10.1126/science.aan5630 (2017).
- 7 Paolucci, C., Di Iorio, J. R., Schneider, W. F. & Gounder, R. Solvation and Mobilization of Copper Active Sites in Zeolites by Ammonia: Consequences for the Catalytic Reduction of Nitrogen Oxides. *Accounts Chem Res* **53**, 1881-1892, doi:10.1021/acs.accounts.0c00328 (2020).
- 8 Ottinger, N., Xi, Y., Keturakis, C. & Liu, Z. Impact of Water Vapor on the Performance of a Cu-SSZ-13 Catalyst under Simulated Diesel Exhaust Conditions. *SAE Int. J. Adv. & Curr. Prac. in Mobility* **3**, 2872-2877, doi:<https://doi.org/10.4271/2021-01-0577> (2021).
- 9 Jones, C. B. *et al.* Effects of dioxygen pressure on rates of NO_x selective catalytic reduction with NH₃ on Cu-CHA zeolites. *J Catal* **389**, 140-149, doi:10.1016/j.jcat.2020.05.022 (2020).
- 10 Wang, D. *et al.* Selective Catalytic Reduction of NO_x with NH₃ over a Cu-SSZ-13 Catalyst Prepared by a Solid-State Ion-Exchange Method. *Chemcatchem* **6**, 1579-1583, doi:DOI 10.1002/cctc.201402010 (2014).
- 11 Bruzzese, P. C. *et al.* O-17-EPR determination of the structure and dynamics of copper single-metal sites in zeolites. *Nat Commun* **12**, 4638, doi:ARTN 4638 10.1038/s41467-021-24935-7 (2021).
- 12 Negri, C. *et al.* Evidence of Mixed-Ligand Complexes in Cu-CHA by Reaction of Cu Nitrates with NO/NH₃ at Low Temperature. *Chemcatchem* **11**, 3828-3838, doi:10.1002/cctc.201900590 (2019).

- 13 Marberger, A. *et al.* Time-resolved copper speciation during selective catalytic reduction of NO on Cu-SSZ-13. *Nature Catalysis* **1**, 221-227, doi:10.1038/s41929-018-0032-6 (2018).
- 14 Bruzzese, P. C. *et al.* The Structure of Monomeric Hydroxo-C-II Species in Cu-CHA. A Quantitative Assessment. *J Am Chem Soc*, doi:10.1021/jacs.2c06037 (2022).
- 15 Paolucci, C. *et al.* Catalysis in a Cage: Condition-Dependent Speciation and Dynamics of Exchanged Cu Cations in SSZ-13 Zeolites. *J Am Chem Soc* **138**, 6028-6048, doi:10.1021/jacs.6b02651 (2016).
- 16 Godiksen, A. *et al.* Coordination Environment of Copper Sites in Cu-CHA Zeolite Investigated by Electron Paramagnetic Resonance. *J Phys Chem C* **118**, 23126-23138, doi:10.1021/jp5065616 (2014).
- 17 Zhang, Y. N. *et al.* Selective Catalytic Reduction of NO_x with NH₃ over Cu/SSZ-13: Elucidating Dynamics of Cu Active Sites with In Situ UV-Vis Spectroscopy and DFT Calculations. *J Phys Chem C* **126**, 8720-8733, doi:10.1021/acs.jpcc.2c01268 (2022).
- 18 Sushkevich, V. L., Smimov, A. V. & van Bokhoven, J. A. Autoreduction of Copper in Zeolites: Role of Topology, Si/Al Ratio, and Copper Loading. *J Phys Chem C* **123**, 9926-9934, doi:10.1021/acs.jpcc.9b00986 (2019).
- 19 Vierke, G. Electron Paramagnetic Resonance Study of Copper(II)-Tetrammine Nitrate in Solution. *Z Naturforsch Pt A* **A 26**, 554-&, doi:DOI 10.1515/zna-1971-0326 (1971).
- 20 Flentge, D. R., Lunsford, J. H., Jacobs, P. A. & Uytterhoeven, J. B. Spectroscopic Evidence for Tetraamminecopper(II) Complex in a Y-Type Zeolite. *J Phys Chem-Us* **79**, 354-360, doi:DOI 10.1021/j100571a014 (1975).

REVIEWER COMMENTS

Reviewer #1 (Remarks to the Author):

Based on their rebuttal, I have two more remarks to the authors. I think they should be addressed before the paper can be accepted for publication in Nature Communications.

1. On the relationship between steady-state and transient kinetics of SCR reactions.

“transient NH₃+NO titration kinetic analysis is not fully consistent with kinetic behavior derived from steady-state measurements where over a wide range of Cu density, SCR rates increase linearly with Cu loading (i.e., invariant turnover rates)^{5,6}. We can understand this discrepancy, at least qualitatively, from the notion that transient NH₃+NO titration works on all reducible Cu(I)-ions irrespective of their relative reactivity, but steady-state kinetics are likely contributed most by Cu(I)-ions that are the most reactive. Since Cu-ions in Cu/CHA follow non-mean field kinetic behavior⁷, global kinetic parameters derived by titrating all Cu-ions certainly have their limitations.”

a) The non-mean field argument was invoked by the Purdue group with reference to OHC kinetics, i.e., to rationalize the observation that O₂ alone at 473 K was unable to reoxidize all Cu(I) ions according to their XAS results. Concerning RHC, however, there is substantial evidence from kinetic and operando spectroscopy studies that all Cu(II) ions are reduced by NO + NH₃. So, why should we expect a non-mean field kinetic behavior of the Cu(II) ions in the RHC?

b) Both the Milano and the Cummins + ORNL groups have shown that 2nd order kinetic fits of RHC transients are coherent with the steady-state SCR kinetics and describe equally well the Cu reduction transients starting either from 100% oxidized conditions or from Standard SCR steady-state conditions: in other words, there is no evidence of more or less reactive Cu(II) ions, at least from their data. Then, what are exactly the “most reactive Cu(II) ions” which would contribute to steady-state kinetics according to the authors?

c) Any steady-state reaction mechanism should be able to explain the transient behaviors, too. How does the authors' proposal account for the observed transient redox features?

2. On the effect of BAS on OHC.

“.....it is anticipated that the BAS influence will be less significant since BAS mainly affects RHC”

Can the authors please provide supporting evidence for this claim?

Reviewer #2 (Remarks to the Author):

The Authors have revised the manuscript addressing the comments of the reviewers. The inclusion of the new discussion/data constitute an improvement over the previous version.

There are still however a few issues to consider before the manuscript can be considered for publication:

1) The Authors in their reply to the comment concerning the different Si/Al and Cu/Al ratios for the tested samples, give a reasonable explanation, including new data. However, this explanation, which will be very useful for the general audience is missing in the final manuscript. This discussion should be brought in the main text explaining carefully the reasons behind the choice of the different compositions and the limits and constraints imposed by the systems chosen by the Authors for their study. The results on the new samples the Authors refer to in the reply (Table 1) should be at least mentioned even if they will be published elsewhere in future.

2) The CuII quantitation provided by the Authors in their reply (Table 1), Table 2 in SI and Figure 13SI need to be clarified. No error bars are given. Quantitative EPR is very delicate. Especially for operando studies a full explanation of the adopted procedure needs to be given, including uncertainties.

3) Concerning the determination of the spin-Hamiltonian parameters of the various Cu species, which is done qualitatively in the manuscript, the Authors state that “Operando EPR during SCR, unfortunately, is much more complex to deal with via spectrum simulations.” and add that “In attempting to simulate our operando spectra that contain both N and OL ligands, we did not obtain any meaningful results.” While the intricacies of operando epr can be understood, the Authors should clearly state these limitations acknowledging that with respect to other studies performed under controlled conditions and aimed at model studies, the level of details is limited.

4) The question of the EPR silent $[\text{Cu}(\text{OH})]^{+}$ species needs to be clarified. The pseudo Jahn-Teller effect was invoked in Ref 29 of the manuscript as a possible qualitative explanation for the loss of EPR intensity of highly Cu loaded samples upon dehydration. Jahn Teller effects manifest in EPR through specific spectral features (line-width changes, g-factor averaging) that can be observed in different temperature ranges. While electronic level degeneracy can be a source of fast relaxation or exceedingly large broadening of EPR spectra recorded at high temperature, low temperature spectra reflect the lifting of the energy level degeneracy associated to structural distortions. This is well documented in case of several Cu complexes that can be found in the literature. This is also what is reported in J. Am. Chem. Soc. 2022, 144, 29, 13079–13083. It was perfectly legitimate to invoke the pseudo JT effect in 2014 ([dx.doi.org/10.1021/jp5065616](https://doi.org/10.1021/jp5065616)), however it should be noted that in that same paper the Authors aptly say concerning the particular geometry leading to the postulated pseudo JT effect that “A stringent

treatment of this particular geometry for Cu^{2+} is to our knowledge not found in the literature and is beyond the scope of this work. The following discussion will therefore be qualitative” To date no further proofs for this effect has been provided. Therefore this interpretation appears outdated and keeping in invoking it does no good service to EPR and to science in general.

Attention should concentrate on dimerization or other mechanisms (indicated by the Authors), which appear to be the ultimate responsible for the loss of EPR intensity upon dehydration and heating of zeolites. The difficulty in the EPR approach is therefore that even though now the spectroscopic features of $[\text{Cu}(\text{OH})]^{+}$ are established, it may be not easy to use these data in the case of highly Cu doped samples, or displaying different structural morphologies. This however, should not induce to indulge in “fuzzy” explanations.

5) Pag.8 and Figure 12 in SI. The comment on the “superhyperfine” structure raised in the first run is not properly addressed. What the Authors mean by “spikes” in the high-field region of the spectrum is unclear. I agree the spectra appear to be better resolved and that this better resolution can be associated to a lower mobility. I do not agree in assigning these features to “superhyperfine” interactions. These can simply be explained in term of resolved Cu hyperfine splittings, off-axis turning points of the Cu powder spectrum or multiple species. The presence of superhyperfine interactions need to be demonstrated via simulation of the spectra and/or isotopic substitution or by more advanced experiments. I strongly suggest the Authors to avoid overinterpretation of their data, limiting to stress that the increased resolution in the g perpendicular region of the spectrum can be taken as an indication of a restricted mobility.

The second derivative spectra presented in the reply and Figure 12 of SI are of too low quality and do not add any information. Such low-quality spectra should not be presented in a high ranking journal such as Nat Commun.

Reviewer #1 (Remarks to the Author):

Based on their rebuttal, I have two more remarks to the authors. I think they should be addressed before the paper can be accepted for publication in Nature Communications.

1. On the relationship between steady-state and transient kinetics of SCR reactions.

“transient NH₃+NO titration kinetic analysis is not fully consistent with kinetic behavior derived from steady-state measurements where over a wide range of Cu density, SCR rates increase linearly with Cu loading (i.e., invariant turnover rates)^{5,6}. We can understand this discrepancy, at least qualitatively, from the notion that transient NH₃+NO titration works on all reducible Cu^I-ions irrespective of their relative reactivity, but steady-state kinetics are likely contributed most by Cu^I-ions that are the most reactive. Since Cu-ions in Cu/CHA follow non-mean field kinetic behavior⁷, global kinetic parameters derived by titrating all Cu-ions certainly have their limitations.”

a) The non-mean field argument was invoked by the Purdue group with reference to OHC kinetics, i.e., to rationalize the observation that O₂ alone at 473 K was unable to reoxidize all Cu(I) ions according to their XAS results. Concerning RHC, however, there is substantial evidence from kinetic and operando spectroscopy studies that all Cu(II) ions are reduced by NO + NH₃. So, why should we expect a non-mean field kinetic behavior of the Cu(II) ions in the RHC?

b) Both the Milano and the Cummins + ORNL groups have shown that 2nd order kinetic fits of RHC transients are coherent with the steady-state SCR kinetics and describe equally well the Cu reduction transients starting either from 100% oxidized conditions or from Standard SCR steady-state conditions: in other words, there is no evidence of more or less reactive Cu(II) ions, at least from their data. Then, what are exactly the “most reactive Cu(II) ions” which would contribute to steady-state kinetics according to the authors?

We thank the reviewer for the great comment. We would like to reply a) and b) collectively since they are correlated. First, we would like to apologize for using the term “most reactive Cu(II) ions”, which can be misleading. The same Cu(II) ions, for example Cu^{II}(NH₃)₄, can have very different reactivity in different catalysts. As such, “Cu atomic efficiency” appears a better term in describing reactivity difference of Cu-ions.¹ Second, we would like to point out that the catalysts used in studies by the Milano and the Cummins + ORNL groups are state-of-the-art industrial catalysts with optimized SCR performance. Such catalysts contain primarily redox active ZCu^{II}OH, while redox resistant Z₂Cu^{II} contents are relatively low.²⁻ As such, these catalysts poorly represent a Z₂Cu^{II} dominant scenario, e.g., catalysts with low Si/Al ratios, or catalysts experience harsh hydrothermal aging. In these latter cases, if Z₂Cu^{II} does not hydrolyze to ZCu^{II}OH fancily, then this hydrolysis chemistry (Z₂Cu^{II} + H₂O = ZCu^{II}OH + ZH⁺) will slow down SCR. Thirdly, “all Cu(II) ions are reduced by NO + NH₃” does not necessarily mean that all Cu(II) ions are reduced with equal ease. Again, when the Z₂Cu^{II} hydrolysis chemistry is kinetically hindered, most certainly all Cu(II) ions are not equally reduced. Next, we use our unpublished NH₃+NO and NO+O₂ titration results to demonstrate this argument.

We carried out NH₃+NO and NO+O₂ titration experiments at 200 °C under in situ EPR conditions, in a similar way as the Milano group did for their transient kinetic studies. We first load a catalyst in our EPR tube, raise the temperature to 200 °C under an inert flow. Following that, we introduce NH₃+NO to reduce Cu^{II}-ions and follow the process with continuous EPR scans. To the point where there is no

obvious Cu(II) reduction any longer, we switch the NH_3 flow off and the O_2 flow on to reoxidize Cu^{I} -ions, until EPR signal intensity does not increase within reasonable time duration. Using our Cu/CHA-a catalyst as an example, Fig. 1a presents spectra collected during NH_3+NO titration. Note that each EPR scan takes ~ 90 seconds. The first 13 scans (not shown) were collected under the inert flow; NH_3+NO titration started at scan No. 14, and ended at scan No. 30 (total reaction time ~ 24 min). Fig. 1b presents spectra collected during the subsequent $\text{NO}+\text{O}_2$ titration.

Fig. 1: (a) NH_3+NO and (b) $\text{NO}+\text{O}_2$ titration measurements at 200 °C over Cu/CHA-a. The reactant feed contains 350 ppm NO_x (including ~ 10 ppm NO_2), 350 ppm NH_3 , 10% O_2 , and balanced N_2 at a gas hourly space velocity (GHSV) of $\sim 4 \times 10^5 \text{ h}^{-1}$.

We thereafter integrated EPR visible Cu(II) contents during such titration measurements, and ratioed against that prior to NH_3+NO introduction. The results are shown in Fig. 2.

Fig. 2: EPR visible Cu(II) versus total isolated Cu(II) of the catalysts during NH_3+NO and $\text{NO}+\text{O}_2$ titration measurements at 200 °C over (a) fresh catalysts, and (b) HTA850 catalysts.

From the results shown in Fig. 2a, it is readily seen that the 4 fresh catalysts respond to NH_3+NO and $\text{NO}+\text{O}_2$ titrations very differently. During NH_3+NO reduction, small quantities of Cu^{I} -ions (presumably in

the form of $\text{Cu}^{\text{II}}(\text{NH}_3)_4$ in Cu/CHA-a, b and Cu/AEI, and large quantities of Cu^{II} -ions in Cu/LTA cannot be readily reduced. Similarly, during the subsequent $\text{NO}+\text{O}_2$ oxidation, large quantities of Cu^{I} -ions in Cu/AEI and Cu/CHA-b are not readily oxidized. For the hydrothermally aged catalysts shown in Fig. 2b, since Cu^{II} -ions in these catalysts mainly stay as $\text{Z}_2\text{Cu}^{\text{II}}$ and residual Brønsted acid density is now rather low due to support dealumination, large quantities $\text{Cu}^{\text{II}}(\text{NH}_3)_4$ become catalytically inert; they do not respond to NH_3+NO reduction at 200 °C within reasonable reaction time.

These new data shown in Figs. 1 and 2 suggest that the non-mean field argument goes beyond OHC by O_2 alone. Even though Cu-CHA catalysts with relatively high Cu loadings and dominant $\text{ZCu}^{\text{II}}\text{OH}$ speciation (e.g., commercial catalysts and Cu/CHA-a used here) may not display strong non-mean field kinetic behavior, this behavior is clearly not universal. When $\text{Z}_2\text{Cu}^{\text{II}}$ hydrolysis and intercage transfer are hindered, the kinetic consequences are obvious. This has been one of the key conclusions of the present study.

c) Any steady-state reaction mechanism should be able to explain the transient behaviors, too. How does the authors' proposal account for the observed transient redox features?

We do not necessarily agree with the argument that “Any steady-state reaction mechanism should be able to explain the transient behaviors”. Our rationale lies in the fact that NH_3 functions both as a reactant and a solvation agent during low temperature SCR. As such, the “quasi homogeneous” scenario is only maintained when gas phase NH_3 is present. For example, OHC during steady-state SCR is different from OHC under transient $\text{NO}+\text{O}_2$ titration, where the former regenerates NH_3 -solved Cu^{II} -ions and the latter generates lattice oxygen coordinated Cu^{II} -ions.

Again, we argue here that transient redox features observed on catalysts with relatively high Cu contents and high $\text{ZCu}^{\text{II}}\text{OH}$ speciation do not capture potential kinetic hinderance from $\text{Z}_2\text{Cu}^{\text{II}}$ hydrolysis and intercage transfer.

2. On the effect of BAS on OHC.

“.....it is anticipated that the BAS influence will be less significant since BAS mainly affects RHC”

Can the authors please provide supporting evidence for this claim?

It is our understanding that O_2 activation by a pair of $\text{Cu}^{\text{I}}(\text{NH}_3)_2$ during OHC is energetically less demanding than the formation of NH_2NO or NH_4NO_2 intermediates during RHC, as suggested by prior theoretical studies.^{5, 6} As such, strong OHC rate-limiting only occurs when Cu loadings are low, or O_2 supply is limited. Low Cu loading or low O_2 pressure are obviously not strongly associated with BAS. RHC kinetics, on the other hand, can be hindered by $\text{Z}_2\text{Cu}^{\text{II}}$ hydrolysis and intercage transfer, which, according to our DFT calculations shown in the present study, are both directly correlated with BAS.

To collectively address these two comments, on page 10 of the revised manuscript, the last paragraph before the Conclusion has been extensively modified, as follows:

Finally, regarding the interplay between RHC and OHC in low-temperature SCR, recent transient kinetic studies by researchers from Milano^{4, 7} and Cummins³ made important discoveries by isolating RHC and OHC with NH_3+NO and $\text{NO}+\text{O}_2$ titrations, respectively, and then combining the two to generate the overall redox kinetic model. In these studies, the authors used catalysts

from major industrial SCR catalyst suppliers (BASF, Johnson-Matthey Inc.). We note that such catalysts are performance optimized; their redox active $ZCu^{II}OH$ contents are typically high, and redox resistant Z_2Cu^{II} contents are typically low. Furthermore, Z_2Cu^{II} hydrolysis to $ZCu^{II}OH$ under SCR conditions appears to be facile for these catalysts⁸. As such, catalytic function of all Cu-ions can be considered equal (e.g., all Cu^{II} -ions are readily reduced during the RHC step⁷), and a mean-field kinetic model works well for describing OHC³. In addition to transient kinetic studies, the interplay between RHC and OHC can also be systematically probed under steady-state conditions using catalysts with a wide range of active Cu densities, and/or O_2 partial pressures^{1, 5, 9, 10}, where OHC rate-controlling can be purposely designed, e.g., by dramatically lowering Cu concentration¹, or by lowering O_2 pressure during SCR reaction¹⁰. In some of such studies, non-mean-field behavior was found to better describe OHC⁵. More generally, these latter studies demonstrate that the mechanisms of RHC and OHC do not appear to change with Cu density; rather the relative intrinsic rates of these processes change with reaction conditions, and neither one solely limits the rates of the SCR redox cycle^{1, 10}. In the present study, we show that support BAS strength and density play important roles on RHC rates. These new findings broaden our understanding on RHC rate-controlling by incorporating support topology and hydrothermal aging effects. Because of the EPR silent nature of Cu^I -ions and dimeric Cu intermediates that are involved in OHC, *operando* EPR cannot be applied to address this half-cycle. However, since BAS mainly influences hydrolysis and intercage transfer of Cu^{II} -ions, it is anticipated that the BAS influence will be less significant roles to OHC. This tentative conclusion, however, awaits additional experimental and theoretical support.

Reviewer #2 (Remarks to the Author):

The Authors have revised the manuscript addressing the comments of the reviewers. The inclusion of the new discussion/data constitute an improvement over the previous version.

We appreciate the reviewer's positive remarks on improvement we made during revision.

There are still however a few issues to consider before the manuscript can be considered for publication:

1) The Authors in their reply to the comment concerning the different Si/Al and Cu/Al ratios for the tested samples, give a reasonable explanation, including new data. However, this explanation, which will be very useful for the general audience is missing in the final manuscript. This discussion should be brought in the main text explaining carefully the reasons behind the choice of the different compositions and the limits and constraints imposed by the systems chosen by the Authors for their study. The results on the new samples the Authors refer to in the reply (Table 1) should be at least mentioned even if they will be published elsewhere in future.

We thank the reviewer for this comment.

On page 2 of the revised manuscript, the following description has been added that reads "Under the current state-of-the-art for small pore zeolite synthesis, CHA can be synthesized with a wide range of Si/Al ratios, however, it is not yet the case for LTA and AEI. Therefore, we prepared CHA-a and CHA-b supports to match Si/Al ratios of our LTA and AEI supports, respectively.

Regarding Cu loading, we used aqueous solution ion exchange to obtain catalysts with intermediate Cu/Al ratios to avoid the presence of multinuclear Cu moieties to ease our kinetics and EPR studies. We believe that such a catalyst preparation approach enables fair comparisons among the 4 catalysts.....”

On page 3 of the revised manuscript, the following sentence has been added: “Using CHA-a and LTA as supports, we also synthesized two series of catalysts with identical Cu loadings via solid-state ion exchange, and then compared their SCR performance. Their Cu atomic efficiency trend fully reproduced the results shown here. The results will be published elsewhere.”

2) The Cu^{II} quantitation provided by the Authors in their reply (Table 1), Table 2 in SI and Figure 13SI need to be clarified. No error bars are given. Quantitative EPR is very delicate. Especially for operando studies a full explanation of the adopted procedure needs to be given, including uncertainties.

Our detailed Cu(II) quantification procedures can be found elsewhere.¹¹ In short, we double-integrate the EPR spectra to obtain signal areas, which are proportional to EPR-active isolated Cu^{II}-ion content. To quantify these, a series of standard solutions with different isolated Cu^{II}-ion concentrations were prepared by dissolving Cu(NO₃)₂·2.5H₂O and imidazole (Sigma Aldrich, 99.0%) in ethylene glycol (Sigma Aldrich, 99.8%). Imidazole was used here to coordinate with Cu^{II}-ions to prevent formation of EPR-silent multinuclear Cu species. The linear calibration curve generated using the integral areas and Cu content of the standard solutions was then used for quantification of isolated Cu^{II} in the catalysts.

The errors for EPR quantification are typically within 5% according to the general wisdom. For ex situ measurements, we typically run at LN₂ temperature to avoid signal loss due to Cu mobility. The accuracy for Cu quantification can be readily confirmed by measurements via other techniques, e.g., ICP analysis, or H₂ temperature-programmed reduction (H₂-TPR). For example, we used both EPR and H₂-TPR in the past to quantify isolated Cu^{II}-ions in a series of Cu/H-SSZ-13 samples, and confirmed good accuracy of our EPR quantification, shown in Fig. 3 below.¹²

Fig. 3 Concentrations of EPR active Cu(II)-ions versus high-temperature (> 700 °C) H₂ consumption in H₂-TPR for Cu/H-SSZ-13 samples.

For operando measurements, we also measure peak areas and then convert to Cu(II) concentrations. An additional correction to make here is the temperature effect: as EPR measurements conducted at

different temperatures follow a Boltzmann distribution,¹³ this temperature correction was also incorporated during our quantification. Regarding uncertainties in operando measurements, we can only confirm that for measurements conducted at 50 °C (below SCR onset temperature), Cu(II) quantification perfectly matches ex situ measurement on hydrated sample at LN2 temperature. For measurements carried out above SCR onset temperature, quantification uncertainties are difficult to estimate since it is not possible to isolate signal loss from Cu(I) formation, from high Cu(II) mobility, and from measurement error.

On page 13 of revised manuscript (method section), the following descriptions have been added to address this comment:

.....Good accuracy for EPR quantification (typical uncertainties within 5%) has been verified by Cu-ion quantification via other techniques, e.g., H₂ temperature-programmed reduction (H₂-TPR)¹².....

.....To quantify EPR active Cu^{II} under operando conditions, the same spectrum double-integration method described above was adopted. An additional correction was made here: as EPR measurements conducted at different temperatures follow a Boltzmann distribution (i.e., peak areas are inversely proportional to the measurement temperature in kelvin)¹³, this temperature correction was also incorporated during our quantification. For measurements carried out above SCR onset temperatures, however, quantification uncertainties are difficult to estimate since it is not possible to isolate signal loss from Cu^I-ion formation, from high Cu^{II}-ion mobility, and from measurement error.

3) Concerning the determination of the spin-Hamiltonian parameters of the various Cu species, which is done qualitatively in the manuscript, the Authors state that "Operando EPR during SCR, unfortunately, is much more complex to deal with via spectrum simulations." and add that "In attempting to simulate our operando spectra that contain both N and OL ligands, we did not obtain any meaningful results." While the intricacies of operando EPR can be understood, the Authors should clearly state these limitations acknowledging that with respect to other studies performed under controlled conditions and aimed at model studies, the level of details is limited.

The reviewer's comment is well received. Some other operando techniques, in particular XANES, use spectra of model Cu species (e.g., Cu^{II}(H₂O)₆, Cu^{II}(NH₃)₄, Cu^I(NH₃)₂, Z-Cu^{II}) and linear combination fit to derive concentration-temperature correlations of such species.¹⁴⁻¹⁶ As for operando EPR, the spin-Hamiltonian parameters of the various Cu species vary much more dramatically as a function of the nature of the ligands (N or O), and temperature. As such, detailed quantitative description of Cu species via spectrum simulation, like in the case of operando XAS, is difficult. Another limitation, i.e., Cu(II) ions are not always visible during operando operation that the reviewer raised during the first round revision, should also be better addressed.

On page 6 of the revised manuscript, we clearly described these limitations, and provided a tentative solution to such limitations. Key modifications are shown below:

.....In recent operando X-ray absorption spectroscopic (XAS) studies for SCR, low-temperature spectra were typically simulated invoking three Cu moieties, namely mobile Cu^{II} (m-Cu^{II}, Cu^{II}(NH₃)₄), mobile Cu^I (m-Cu^I, Cu^I(NH₃)₂), and zeolite-bound Cu^{II} (Z-Cu^{II})^{14, 16, 17}. In the present study, the EPR invisibility of mobile Cu^{II}(NH₃)₄ on Cu/CHA and Cu/AEI at 150/225 °C,

and the visibility of the same species on Cu/LTA at the same temperatures, therefore, reveal an important difference between Cu/LTA and the other three catalysts, that is, $\text{Cu}^{\text{II}}(\text{NH}_3)_4$ likely undergoes very rapid solvation-desolvation (i.e., $\text{Cu}^{\text{II}}(\text{NH}_3)_4 \rightleftharpoons \text{NH}_3 + \text{Cu}^{\text{II}}(\text{NH}_3)_{4-x}(\text{O}_L)_x$) on CHA and AEI, rendering detectability of immobilized $\text{Cu}^{\text{II}}(\text{NH}_3)_{4-x}(\text{O}_L)_x$ alone. In contrast, this rapid interconversion does not appear to establish on LTA, rendering the detectability of primarily mobile $\text{Cu}^{\text{II}}(\text{NH}_3)_4$. In the following, DFT calculations will be used to further elucidate such a dramatic support effect.

.....Note that in using quantitative EPR measurements to correlate Cu atomic efficiency, one limitation is that some Cu^{II} moieties may become EPR silent due, for example, to fast relaxations. As discussed above, rapid interactions with the supports render $\text{Cu}^{\text{II}}(\text{NH}_3)_4$ EPR invisible on CHA and AEI at 150/225 °C, even though presence of $\text{Cu}^{\text{II}}(\text{NH}_3)_4$ has been repeatedly confirmed by operando XAS studies under similar conditions^{14, 16, 17}. Fortunately, the EPR visibility and Cu atomic efficiency correlation appears to hold here even with this uncertainty. Another limitation for quantitative operando EPR, as indicated by spectra shown in Fig. 3, is that spin-Hamiltonian parameters for Cu species vary rather dramatically with the nature of the ligands (N or O), and with temperature. As such, detailed quantitative description of Cu species via spectrum simulation using linear combination fit of model species as in the case of operando XAS, is not yet achievable. However, spin-Hamiltonian parameters of SCR relevant model species, e.g., $\text{Cu}^{\text{II}}(\text{NH}_3)_4$, $\text{Cu}^{\text{II}}(\text{OH})(\text{NH}_3)_3$, or even the $[\text{Cu}^{\text{II}}(\text{NH}_3)_3(\text{NO}_3)]^+$ complex first prepared by Negri et al.¹⁸, can be readily measured at cryogenic temperatures. In this case, EPR spectra acquired on working SCR catalysts rapidly quenched to the same temperatures may be simulated using spin-Hamiltonian parameters of such model Cu species.

4) The question of the EPR silent $[\text{Cu}(\text{OH})]^+$ species needs to be clarified. The pseudo Jahn-Teller effect was invoked in Ref 29 of the manuscript as a possible qualitative explanation for the loss of EPR intensity of highly Cu loaded samples upon dehydration. Jahn Teller effects manifest in EPR through specific spectral features (line-width changes, g-factor averaging) that can be observed in different temperature ranges. While electronic level degeneracy can be a source of fast relaxation or exceedingly large broadening of EPR spectra recorded at high temperature, low temperature spectra reflect the lifting of the energy level degeneracy associated to structural distortions. This is well documented in case of several Cu complexes that can be found in the literature. This is also what is reported in *J. Am. Chem. Soc.* 2022, 144, 29, 13079–13083. It was perfectly legitimate to invoke the pseudo JT effect in 2014 ([dx.doi.org/10.1021/jp5065616](https://doi.org/10.1021/jp5065616)), however it should be noted that in that same paper the Authors aptly say concerning the particular geometry leading to the postulated pseudo JT effect that “A stringent treatment of this particular geometry for Cu^{2+} is to our knowledge not found in the literature and is beyond the scope of this work. The following discussion will therefore be qualitative” To date no further proofs for this effect has been provided. Therefore this interpretation appears outdated and keeping in invoking it does no good service to EPR and to science in general. Attention should concentrate on dimerization or other mechanisms (indicated by the Authors), which appear to be the ultimate responsible for the loss of EPR intensity upon dehydration and heating of zeolites. The difficulty in the EPR approach is therefore that even though now the spectroscopic features of $[\text{Cu}(\text{OH})]^+$ are established, it may be not easy to use these data in the case of highly Cu doped

samples, or displaying different structural morphologies. This however, should not induce to indulge in “fuzzy” explanations.

We are grateful to the reviewer for the detailed description of the unlikelihood for the pseudo JT effect to apply. In agreement with the reviewer, on page 4 of the revised manuscript, the following modifications have been made:

.....Since such hyperfine signals are barely detected in our dehydrated samples, the loss of $ZCu^{II}OH$ signals is likely due to other causes. From recent literature, $ZCu^{II}OH$ has been known to lose EPR visibility during dehydration via dimerization and autoreduction chemistries^{19, 20}. As such, we assign all ESR signals in Fig. 2c to Z_2Cu^{II} , and tentatively suggest that the loss of EPR signal is due to $ZCu^{II}OH$ conversion to EPR silent moieties.

Despite this modification, which we hope the reviewer can accept, we still have some concerns regarding the atomic structure of $ZCu^{II}OH$ proposed in the recent study by Chiesa and coworkers²¹, which is shown below (this notion is to share with the reviewer, and not for the purpose of revision).

Fig. 4 (a) atomic structure of $ZCu^{II}OH$ derived in Ref²¹, (b) atomic structure of $ZCu^{II}OH$ derived from prior XAS and DFT simulations¹³.

Prior to this recent publication that suggests that $ZCu^{II}OH$ has a Cu-O coordination number of 4, it is generally agreed, based on XAS and DFT, that dehydrated $ZCu^{II}OH$ has a Cu-O coordination number of 3 (Fig. 3b). This made us suspect that the EPR active “ $ZCu^{II}OH$ ” that Chiesa and coworkers observed is actually not a fully dehydrated species, but a partially hydrated $ZCu^{II}(H_2O)OH$ – its EPR activity is due to 4 Cu-O coordination, but different from what they proposed. From our experience, it is very common to have traces of H_2O in UHP carrier gases that can readily hydrate zeolite materials.

5) Pag.8 and Figure 12 in SI. The comment on the “superhyperfine” structure raised in the first run is not properly addressed. What the Authors mean by “spikes” in the high-field region of the spectrum is unclear. I agree the spectra appear to be better resolved and that this better resolution can be associated to a lower mobility. I do not agree in assigning these features to “superhyperfine” interactions. These can simply be explained in term of resolved Cu hyperfine splittings, off-axis turning points of the Cu powder spectrum or multiple species. The presence of superhyperfine interactions need

to be demonstrated via simulation of the spectra and/or isotopic substitution or by more advanced experiments. I strongly suggest the Authors to avoid overinterpretation of their data, limiting to stress that the increased resolution in the g_{\perp} perpendicular region of the spectrum can be taken as an indication of a restricted mobility.

The second derivative spectra presented in the reply and Figure 12 of SI are of too low quality and do not add any information. Such low-quality spectra should not be presented in a high ranking journal such as Nat Commun.

We thank the reviewer for the great comment. We observe two types of features that we call “spikes” in the g_{\perp} region. For type I, an example is shown below that should due to $\text{Cu}^{\text{II}}(\text{NH}_3)_4$. Fig. 5a displays in situ EPR spectra for Cu/CHA-b titrated by NH_3+NO at 200 °C (experiment details can be found above in our reply to comments from Reviewer #1). As the titration continues, some residual Cu^{II} -ions become difficult to reduce, and the g_{\perp} region spectra display equally spaced fine structures highlighted in Fig. 5b. We believe that these are indeed due to superhyperfine interactions between Cu and NH_3 in $\text{Cu}^{\text{II}}(\text{NH}_3)_4$ with low mobility.

Fig. 5 (a) in situ EPR spectra acquired during NH_3+NO titration of Cu/CHA-b at 200 °C. (b) enlarged g_{\perp} region of scan No. 20.

For type II, which we observe in spectra shown in, for example Fig. 7c of the main text, may indeed be due to other causes suggested by the reviewer, e.g., resolved Cu hyperfine splittings, off-axis turning points of the Cu powder spectrum or multiple species.

In agreement with the reviewer, we deleted the descriptions regarding superhyperfine signals, and removed Fig. S12 that we added to the SI file during our first-round revision.

References:

- (1) Wu, Y. Q.; Ma, Y.; Wang, Y. L.; Rappe, K. G.; Washton, N. M.; Wang, Y.; Walter, E. D.; Gao, F. Rate Controlling in Low-Temperature Standard NH₃-SCR: Implications from Operando EPR Spectroscopy and Reaction Kinetics. *J Am Chem Soc* **2022**, *144* (22), 9734-9746. DOI: 10.1021/jacs.2c01933.
- (2) Daya, R.; Trandal, D.; Dadi, R. K.; Li, H.; Joshi, S. Y.; Luo, J. Y.; Kumar, A.; Yezerets, A. Kinetics and thermodynamics of ammonia solvation on Z(2)Cu, ZCuOH and ZCu sites in Cu-SSZ-13-Implications for hydrothermal aging. *Appl Catal B-Environ* **2021**, *297*. DOI: ARTN 120444
10.1016/j.apcatb.2021.120444.
- (3) Daya, R.; Trandal, D.; Menon, U.; Deka, D. J.; Partridge, W. P.; Joshi, S. Y. Kinetic Model for the Reduction of Cu(II) Sites by NO + NH₃ and Reoxidation of NH₃-Solvated Cu(I) Sites by O₂ and NO in Cu-SSZ-13. *Acs Catal* **2022**, *12* (11), 6418-6433. DOI: 10.1021/acscatal.2c01076.
- (4) Hu, W. S.; Selleri, T.; Gramigni, F.; Fenes, E.; Rout, K. R.; Liu, S. J.; Nova, I.; Chen, D.; Gao, X.; Tronconi, E. On the Redox Mechanism of Low-Temperature NH₃-SCR over Cu-CHA: A Combined Experimental and Theoretical Study of the Reduction Half Cycle. *Angew Chem Int Edit* **2021**, *60* (13), 7197-7204. DOI: 10.1002/anie.202014926.
- (5) Paolucci, C.; Khurana, I.; Parekh, A. A.; Li, S. C.; Shih, A. J.; Li, H.; Di Iorio, J. R.; Albarracin-Caballero, J. D.; Yezerets, A.; Miller, J. T.; et al. Dynamic multinuclear sites formed by mobilized copper ions in NO_x selective catalytic reduction. *Science* **2017**, *357* (6354), 898-903. DOI: 10.1126/science.aan5630.
- (6) Chen, L.; Falsig, H.; Janssens, T. V. W.; Jansson, J.; Skoglundh, M.; Gronbeck, H. Effect of Al-distribution on oxygen activation over Cu-CHA. *Catal Sci Technol* **2018**, *8* (8), 2131-2136. DOI: 10.1039/c8cy00083b.
- (7) Gramigni, F.; Nasello, N. D.; Usberti, N.; Iacobone, U.; Selleri, T.; Hu, W. S.; Liu, S. J.; Gao, X.; Nova, I.; Tronconi, E. Transient Kinetic Analysis of Low-Temperature NH₃-SCR over Cu-CHA Catalysts Reveals a Quadratic Dependence of Cu Reduction Rates on Cu-II. *Acs Catal* **2021**, *11* (8), 4821-4831. DOI: 10.1021/acscatal.0c05362.
- (8) Hu, W. S.; Iacobone, U.; Gramigni, F.; Zhang, Y.; Wang, X. X.; Liu, S. J.; Zheng, C. H.; Nova, I.; Gao, X.; Tronconi, E. Unraveling the Hydrolysis of Z(2)Cu(2+) to ZCu(2+)(OH)(-) and Its Consequences for the Low-Temperature Selective Catalytic Reduction of NO on Cu-CHA Catalysts. *Acs Catal* **2021**, *11* (18), 11616-11625. DOI: 10.1021/acscatal.1c02761.
- (9) Gao, F.; Mei, D. H.; Wang, Y. L.; Szanyi, J.; Peden, C. H. F. Selective Catalytic Reduction over Cu/SSZ-13: Linking Homo- and Heterogeneous Catalysis. *J Am Chem Soc* **2017**, *139* (13), 4935-4942. DOI: 10.1021/jacs.7b01128.
- (10) Jones, C. B.; Khurana, I.; Krishna, S. H.; Shih, A. J.; Delgass, W. N.; Miller, J. T.; Ribeiro, F. H.; Schneider, W. F.; Gounder, R. Effects of dioxygen pressure on rates of NO_x selective catalytic reduction with NH₃ on Cu-CHA zeolites. *J Catal* **2020**, *389*, 140-149. DOI: 10.1016/j.jcat.2020.05.022.
- (11) Cui, Y. R.; Wang, Y. L.; Mei, D. H.; Walter, E. D.; Washton, N. M.; Holladay, J. D.; Wang, Y.; Szanyi, J.; Peden, C. H. F.; Gao, F. Revisiting effects of alkali metal and alkaline earth co-cation additives to Cu/SSZ-13 selective catalytic reduction catalysts. *J Catal* **2019**, *378*, 363-375. DOI: 10.1016/j.jcat.2019.08.028.
- (12) Cui, Y. R.; Wang, Y. L.; Walter, E. D.; Szanyi, J.; Wang, Y.; Gao, F. Influences of Na⁺ co-cation on the structure and performance of Cu/SSZ-13 selective catalytic reduction catalysts. *Catal Today* **2020**, *339*, 233-240. DOI: 10.1016/j.cattod.2019.02.037.
- (13) Roessler, M. M.; Salvadori, E. Principles and applications of EPR spectroscopy in the chemical sciences. *Chemical Society Reviews* **2018**, *47* (8), 2534-2553. DOI: 10.1039/c6cs00565a.
- (14) Lomachenko, K. A.; Borfecchia, E.; Negri, C.; Berlier, G.; Lamberti, C.; Beato, P.; Falsig, H.; Bordiga, S. The Cu-CHA deNO(x) Catalyst in Action: Temperature-Dependent NH₃-Assisted Selective Catalytic Reduction Monitored by Operando XAS and XES. *J Am Chem Soc* **2016**, *138* (37), 12025-12028. DOI: 10.1021/jacs.6b06809.

- (15) Borfecchia, E.; Negri, C.; Lomachenko, K. A.; Lamberti, C.; Janssens, T. V. W.; Berlier, G. Temperature-dependent dynamics of NH₃-derived Cu species in the Cu-CHA SCR catalyst. *React Chem Eng* **2019**, *4* (6), 1067-1080. DOI: 10.1039/c8re00322j.
- (16) Kubota, H.; Liu, C.; Amada, T.; Kon, K.; Toyao, T.; Maeno, Z.; Ueda, K.; Satsuma, A.; Tsunooji, N.; Sano, T.; et al. In situ/operando spectroscopic studies on NH₃-SCR reactions catalyzed by a phosphorus-modified Cu-CHA zeolite. *Catal Today* **2021**, *376*, 73-80. DOI: 10.1016/j.cattod.2020.07.084.
- (17) Marberger, A.; Petrov, A. W.; Steiger, P.; Elsener, M.; Krocher, O.; Nachttegaal, M.; Ferri, D. Time-resolved copper speciation during selective catalytic reduction of NO on Cu-SSZ-13. *Nature Catalysis* **2018**, *1* (3), 221-227. DOI: 10.1038/s41929-018-0032-6.
- (18) Negri, C.; Borfecchia, E.; Cutini, M.; Lomachenko, K. A.; Janssens, T. V. W.; Berlier, G.; Bordiga, S. Evidence of Mixed-Ligand Complexes in Cu-CHA by Reaction of Cu Nitrates with NO/NH₃ at Low Temperature. *Chemcatchem* **2019**, *11* (16), 3828-3838. DOI: 10.1002/cctc.201900590.
- (19) Zhang, Y. N.; Zhang, J.; Wang, H. L.; Yang, W. N.; Wang, C. Z.; Peng, Y.; Chen, J. J.; Li, J. H.; Gao, F. Selective Catalytic Reduction of NO_x with NH₃ over Cu/SSZ-13: Elucidating Dynamics of Cu Active Sites with In Situ UV-Vis Spectroscopy and DFT Calculations. *J Phys Chem C* **2022**, *126*, 8720-8733. DOI: 10.1021/acs.jpcc.2c01268.
- (20) Sushkevich, V. L.; Smimov, A. V.; van Bokhoven, J. A. Autoreduction of Copper in Zeolites: Role of Topology, Si/Al Ratio, and Copper Loading. *J Phys Chem C* **2019**, *123* (15), 9926-9934. DOI: 10.1021/acs.jpcc.9b00986.
- (21) Bruzzese, P. C.; Salvadori, E.; Civalleri, B.; Jaeger, S.; Hartmann, M.; Poepl, A.; Chiesa, M. The Structure of Monomeric Hydroxo-C-II Species in Cu-CHA. A Quantitative Assessment. *J Am Chem Soc* **2022**. DOI: 10.1021/jacs.2c06037.

REVIEWERS' COMMENTS

Reviewer #1 (Remarks to the Author):

I truly appreciate the authors' efforts in further revising and improving their manuscript.

The key message well evidenced in this paper is the important role of BAS strength and/or Al density in Cu-zeolite catalysts in controlling the RHC of Standard SCR. This is important, as it explains:

- the ranking of SCR turnover rates (Fig. 1) and of Cu reduction extents (Fig. 6 SI) observed at steady state over Cu-zeolite catalysts with different frameworks
- the reduced activity of hydrothermally aged (850°C) catalysts, where dealumination hinders key steps in the RHC, i.e., hydrolysis of $\text{Cu(II)(NH}_3)_4$ to redox active $\text{Cu(II)(OH)(NH}_3)_3$ and their intercage migration.

Here are just a few more comments from my side.

1. It is unfortunate that the two Cu-CHA catalysts tested in this work have not only different Si/Al ratios but also different Cu loadings: this prevents a direct comparison to confirm the positive effect of the Al density for the same topology of the zeolite framework. The authors mention on p. 3 line 120 that they have synthesized a set of Cu-CHA catalysts with identical Cu loadings and compared their SCR performances, finding results which fully reproduce the trends shown in the present manuscript. In my view, at least some of such results deserve to be included in this paper, where they clearly belong. Otherwise, the correlation between Al density (Si/Al) and SCR activity in Cu-CHA remains a little fuzzy.
2. While the correlation between SCR turnover rates and Cu reduction extents is indeed suggestive of RHC playing a significant role in determining the overall SCR rate, it is to be noted also that the catalyst redox state is about 50% (between 30 and 60%) oxidized in the T-range of interest (150 – 225°C, Figure 6 SI): this points out that neither RHC nor OHC is clearly the rate limiting step of Standard SCR at these conditions.
3. The authors expect that the BAS influence on OHC, not investigated here, will be of less importance, since it affects primarily hydrolysis and intercage transfer of Cu(II) ions. However, a very recent publication from Purdue (<https://doi.org/10.1038/s41929-023-00932-5>) seems to suggest the opposite: they report that increasing the Al density in Cu-CHA leads primarily to an increase of the fraction of oxidizable Cu(I) ions and of the Cu(I) oxidation rate constants, while the reduction rate constants exhibit a maximum with Al density. This paper is obviously relevant: it should be cited and discussed.

Reviewer #2 (Remarks to the Author):

I thank the Authors for their kind reply, they have satisfactorily addressed my concerns with additional

discussion, clarification in the main text and simulations.

I only have one final comment, which is not strictly related to the finding of this manuscript but I feel it is due in reply to the Authors concerns regarding the atomic structure of ZCuIIOH recently proposed in the literature. On reading the results reported in 10.1021/jacs.2c06037, it appears that the proposed structure is related to low temperature regimes, while the three coordinated structure mentioned by the Authors and reported in 10.1039/C4SC02907K refers to high temperature experiments (673 K). This is a key point the Authors should pay attention to. On reading SI of 10.1021/jacs.2c06037 they will see that the computed energy difference between the 4c and 3c structures is only 5 kJ/mol, meaning that at high temperature the 3c structure is the likely one. So EPR in the low temperature regime will see the 4c structure, while operando XAS studies at high T see the 3c structure. There is no contradiction here, but simply a different coordination chemistry regime determined by the temperature conditions.

Reviewer #1 (Remarks to the Author):

I truly appreciate the authors' efforts in further revising and improving their manuscript.

The key message well evidenced in this paper is the important role of BAS strength and/or Al density in Cu-zeolite catalysts in controlling the RHC of Standard SCR. This is important, as it explains:

- the ranking of SCR turnover rates (Fig. 1) and of Cu reduction extents (Fig. 6 SI) observed at steady state over Cu-zeolite catalysts with different frameworks
- the reduced activity of hydrothermally aged (850°C) catalysts, where dealumination hinders key steps in the RHC, i.e., hydrolysis of $\text{Cu(II)(NH}_3)_4$ to redox active $\text{Cu(II)(OH)(NH}_3)_3$ and their intercage migration.

We are very grateful to this reviewer for his/her highly valued comments. These comments help us to interpret our research findings in a much-improved way.

Here are just a few more comments from my side.

1. It is unfortunate that the two Cu-CHA catalysts tested in this work have not only different Si/Al ratios but also different Cu loadings: this prevents a direct comparison to confirm the positive effect of the Al density for the same topology of the zeolite framework. The authors mention on p. 3 line 120 that they have synthesized a set of Cu-CHA catalysts with identical Cu loadings and compared their SCR performances, finding results which fully reproduce the trends shown in the present manuscript. In my view, at least some of such results deserve to be included in this paper, where they clearly belong. Otherwise, the correlation between Al density (Si/Al) and SCR activity in Cu-CHA remains a little fuzzy.

We agree with the reviewer that the two Cu-CHA catalysts studied here have clear limitations for direct comparison. On page 3 of revision 2, we mentioned that "Using CHA-a and LTA as supports, we also synthesized two series of catalysts with identical Cu loadings via solid-state ion exchange, and then compared their SCR performance." We are afraid that the reviewer misunderstood this description. These new catalysts allow more fair comparison between Cu-CHA and Cu-LTA at the same Cu loadings, however, they do not address the reviewer's question on correlation between Al density and SCR activity in Cu-CHA.

To address this latter question, the best catalysts to use are a series of Cu-CHA with varying Si/Al ratio but identical Cu content. We do not have such a series of catalysts at present, but we do have another series that can be used to address the Si/Al ratio effect which we will describe below in addressing comment #3.

2. While the correlation between SCR turnover rates and Cu reduction extents is indeed suggestive of RHC playing a significant role in determining the overall SCR rate, it is to be noted also that the catalyst redox state is about 50% (between 30 and 60%) oxidized in the T-range of interest (150 – 225°C, Figure 6 SI): this points out that neither RHC nor OHC is clearly the rate limiting step of Standard SCR at these conditions.

We fully agree with the reviewer. On page 10 of R3, the following sentence has been added that reads: "This notion is again corroborated by our operando EPR quantification data in Supplementary Fig. 6, showing that neither Cu^{II} nor Cu^{I} is in absolute dominance under low-temperature SCR

conditions, suggesting that neither RHC nor OHC is clearly the rate limiting step.”

3. The authors expect that the BAS influence on OHC, not investigated here, will be of less importance, since it affects primarily hydrolysis and interstage transfer of Cu(II) ions. However, a very recent publication from Purdue (<https://doi.org/10.1038/s41929-023-00932-5>) seems to suggest the opposite: they report that increasing the Al density in Cu-CHA leads primarily to an increase of the fraction of oxidizable Cu(I) ions and of the Cu(I) oxidation rate constants, while the reduction rate constants exhibit a maximum with Al density. This paper is obviously relevant: it should be cited and discussed.

We thank the reviewer for providing this new reference. We are actually aware of this publication; we didn't include it as a reference since it is only very recently appeared.

Recently, we prepared a series of Cu-CHA catalysts with varying Si/Al ratios from 6 to 36, and loaded Cu via aqueous ion exchange. The compositions are shown below.

Table 1 Si/Al ratios and Cu loadings of the Cu-CHA catalysts

Catalyst	Si/Al ratio	Cu loading (wt.%) via ICP
Cu-6	6	3.69
Cu-12	12	1.72
Cu-18	18	1.48
Cu-24	24	1.17
Cu-30	30	1.05
Cu-36	36	0.98

Using this series of catalysts, we carried out standard SCR. Some key kinetic results are shown below:

Fig. 1: (left) NO_x conversion as a function of temperature in standard SCR over the Cu-CHA catalysts; (right) turnover rate at 150 °C on a per Cu basis. Test condition: 350 ppm NO, 350 ppm NH₃, 10% O₂, 2.5% H₂O, balance N₂ at 200k h⁻¹.

Since these CHA supports have different exchange capacity, and since aqueous solution ion exchange was used to prepare these catalysts, it is difficult to maintain an identical Cu loading. Still, Cu-12 and Cu-18 can be considered to have similar Cu loadings, and Cu-24 to Cu-36 samples can be considered to have similar Cu loadings. Therefore, they can be used to partly address the review comment.

It is interesting that this series of catalysts display highest Cu atomic efficiency at Si/Al ratio of 12, consistent with the finding by the Gounder group that the reduction rate constants exhibit a maximum at ~ 0.8 Al per d6r (i.e., Si/Al = 14). However, even if we assume here that Cu atomic efficiency difference is due entirely to an Al density effect (which is clearly an oversimplification), this effect leads to TOR differences at best 2-3 times among the catalysts. Fig. 1 of the manuscript shows, in contrast, that BAS strength difference can lead to TOR differences of almost an order of magnitude.

Even though the discussion immediately above supports the argument that BAS influences RHC more than OHC, we feel that our steady-state kinetics data shown above are not ideally suited to address this important open question, and further investigations are needed. As such, we decide to discuss this new reference in a general way in our revision, as follows.

In the revised manuscript, we added this new reference. On page 10, we added the following description: In a very recent publication by Gounder and coworkers⁴⁹, the authors demonstrated that increasing the zeolite support Al density leads to systematic increases in both the fraction of Cu^I ions that are SCR active and OHC rate constants. Based on this new discovery and the current study, BAS influences both the fractions of SCR active Cu^{II} during RHC and the fractions of SCR active Cu^I during OHC. It is anticipated that catalyst composition and the history of use/treatment determine which effect plays more important kinetic roles under a given SCR reaction condition, and this will be further addressed in our future studies.

Reviewer #2 (Remarks to the Author):

I thank the Authors for their kind reply, they have satisfactorily addressed my concerns with additional discussion, clarification in the main text and simulations.

The reviewer's positive remarks are greatly appreciated.

I only have one final comment, which is not strictly related to the finding of this manuscript but I feel it is due in reply to the Authors concerns regarding the atomic structure of ZCuII(OH) recently proposed in the literature. On reading the results reported in 10.1021/jacs.2c06037, it appears that the proposed structure is related to low temperature regimes, while the three coordinated structure mentioned by the Authors and reported in 10.1039/C4SC02907K refers to high temperature experiments (673 K). This is a key point the Authors should pay attention to. On reading SI of 10.1021/jacs.2c06037 they will see that the computed energy difference between the 4c and 3c structures is only 5 kJ/mol, meaning that at high temperature the 3c structure is the likely one. So EPR in the low temperature regime will see the 4c structure, while operando XAS studies at high T see the 3c structure. There is no contradiction here, but simply a different coordination chemistry regime determined by the temperature conditions.

We thank the reviewer for the detailed explanation. On page 4 of the revised manuscript, we added "which adopts a 4 Cu-O coordination at cryogenic temperatures as opposed to a 3 Cu-O coordinated structure typically observed at higher temperatures³¹."